

# Insights from mercury stable isotopes on terrestrial – atmosphere exchange of Hg(0) in the Arctic tundra

Martin Jiskra,[1,2] Jeroen E. Sonke,[1] Yannick Agnan,[3,4] Detlev Helmig,[5] Daniel Obrist[3,6]

[1]Laboratoire Géosciences Environnement Toulouse, CNRS/IRD/Université de Toulouse, Toulouse, 31400, France
[2]Environmental Geosciences, University of Basel, Basel, 4056, Switzerland
[3]Division of Atmospheric Sciences, Desert Research Institute, Reno, 89512, USA
[4]Sorbonne Université, CNRS/EPHE/UMR/METIS, Paris, F-75252, France
[5] Institute of Arctic and Alpine Research (INSTAAR), University of Colorado, Boulder, 80309, USA.
[6] Department of Environmental, Earth, and Atmospheric Sciences, University of Massachusetts, Lowell, 01854, USA

Correspondence to: Martin Jiskra (martin.jiskra@unibas.ch)

**Abstract.** The tundra plays a pivotal role in the Arctic mercury (Hg) cycling by storing atmospheric Hg deposition and shuttling it to the Arctic Ocean. A recent study revealed that 70% of the atmospheric Hg deposition to the tundra occurs by gaseous elemental mercury (GEM or Hg(0)) uptake by vegetation and soils. Processes controlling land – atmosphere exchange of Hg(0) in the Arctic tundra are therefore central, but remain understudied. Here, we combine Hg stable isotope analysis of Hg(0) in the atmosphere, interstitial snow and soil pore air, with Hg(0) flux measurements in a tundra ecosystem at Toolik field station in northern Alaska (USA). In dark winter months, planetary boundary layer (PBL) conditions and Hg(0) concentrations were generally stable throughout the day and small Hg(0) net deposition occurred. In spring, halogen-induced atmospheric mercury depletion events (AMDE's) occurred, with fast re-emission of Hg(0) after AMDE's resulting in net emission fluxes of Hg(0). During the short snow-free growing season in summer, vegetation uptake of atmospheric Hg(0) enhanced atmospheric Hg(0) net deposition to the Arctic tundra. At night, when PBL conditions were stable, ecosystem uptake of atmospheric Hg(0) led to a depletion of atmospheric Hg(0). The night time decline of atmospheric Hg(0) was concomitant with a depletion of lighter Hg(0) isotopes in the atmospheric Hg pool. The enrichment factor, $\varepsilon^{202}Hg = -4.2$ ‰ $\pm 1.0$ ‰ was consistent with the preferential uptake of light Hg(0) isotopes by vegetation. Hg(0) flux measurements indicated a partial re-emission of Hg(0) during daytime, when solar radiation was strongest. Hg(0) concentrations in soil pore air were depleted relative to atmospheric Hg(0) concentrations, concomitant with an enrichment of lighter Hg(0) isotopes in the soil pore air ($\varepsilon^{202}Hg_{soilair-atmosphere} = -1.00$ ‰ ($\pm 0.25$ ‰) and $E^{199}Hg_{soilair-atmosphere} = 0.07$ ‰ ($\pm 0.04$ ‰)). These first Hg stable isotope measurements of Hg(0) in soil pore air are consistent with the fractionation previously observed during Hg(0) oxidation by natural humic acids suggesting abiotic oxidation as a cause for observed soil Hg(0) uptake.

## 1 Introduction

Mercury (Hg) is a high priority pollutant causing neurodevelopmental deficits in children and cardiovascular disease in adults (Budtz-Jørgensen et al., 2000;Roman et al., 2011). Arctic populations are particularly exposed to high Hg levels, despite few local anthropogenic Hg emissions sources, due to their traditional diet consisting of high trophic level seafood (Sheehan et al., 2014). Anthropogenic Hg emissions from mid latitudes reach remote ecosystems, such as the Arctic through long-range transport of gaseous elemental mercury (GEM or Hg(0)) (Douglas et al., 2012). It has long been thought that springtime oxidation of Hg(0) driven by photochemically produced bromine radicals Br• on surface snow, named "atmospheric mercury depletion events" (AMDE), leads to enhanced deposition of divalent mercury Hg(II) in Arctic regions (Steffen et al., 2008). Such Hg(II) deposition to the snowpack during AMDE's, however, can be subject to photoreduction and fast re-emission back into the atmosphere, minimizing the net load of Hg by AMDE's to snow (Douglas et al., 2012;Johnson et al., 2008). Tundra soils play a central role in the Arctic Hg cycling by storing atmospheric Hg deposition from where it can be mobilized and transported to the Arctic Ocean (Obrist et al., 2017;Sonke and Heimburger, 2012;Sonke et al., 2018). Over millennia,





tundra vegetation and soils have drawn down Hg(0) from the atmosphere, resulting in one of the largest pools of Hg (408 – 863 Gg Hg, top 1 m) stored at the Earth's surface (Obrist et al., 2017;Olson et al., 2018;Schuster et al., 2018). Arctic rivers have recently been recognized to deliver large amounts of Hg to the Arctic Ocean (Fisher et al., 2012;Sonke et al., 2018). In aquatic ecosystems, Hg can be transformed to methyl-Hg that bioaccumulates in the aquatic food chain resulting in elevated

Hg concentrations in high trophic level fishes and mammals (Outridge et al., 2009;Douglas et al., 2012).

On a global scale, vegetation uptake of atmospheric Hg(0) represents the dominant pathway of atmospheric Hg deposition to terrestrial surfaces, resulting in strong seasonal variation of atmospheric Hg(0) concentrations with minima in summer when vegetation activity is highest (Jiskra et al., 2018). Direct Hg(0) flux measurements over selected surfaces (e.g., soil, snow, or leaves), however, do not always provide conclusive answers about the direction and magnitude of Hg exchange between

terrestrial ecosystems and the atmosphere. For example, a review of 132 terrestrial-atmosphere Hg(0) flux studies conducted over various surfaces in the last 30 years estimated a wide range of net fluxes in the range of −513 to 1650 Mg a$^{-1}$ (37.5$^{th}$ to 62.5$^{th}$ percentile) (Agnan et al., 2016). In recent years, a number of studies using Hg(0) flux measurements based on micro-metrological methods directly quantified net ecosystem exchange fluxes of Hg(0) over terrestrial ecosystems (i.e., at the ecosystem-level), including atmosphere-vegetation exchange and underlying soil/litter contributions (Lindberg et al.,

1998;Fritsche et al., 2008;Osterwalder et al., 2017). Measurements of multi-level Hg(0) gradients and interstitial snow and soil pore air provided additional constraints on the terrestrial surface exchange flux (Sigler and Lee, 2006;Moore and Castro, 2012;Fain et al., 2013;Obrist et al., 2014;Fu et al., 2016b;Agnan et al., 2018).

Hg stable isotopes are a powerful tool to study the deposition and re-emission pathways of Hg to terrestrial ecosystems. The Hg stable isotope fingerprint of soil samples reflects the source contribution of atmospheric Hg(0) dry deposition, Hg(II) wet

deposition, and Hg from geogenic origin as well as processes fractionating Hg isotopes during post-deposition processes, e.g., re-emission (Demers et al., 2013;Jiskra et al., 2015;Enrico et al., 2016). Mercury has seven stable isotopes, which can undergo mass dependent fractionation (MDF, described by $\delta^{202}$Hg), mass-independent fractionation of odd-mass-number isotopes (odd-MIF, described by $\Delta^{199}$Hg and $\Delta^{201}$Hg), and even-mass-number isotopes (even-MIF, described by $\Delta^{200}$Hg and $\Delta^{204}$Hg), thereby producing a multi-dimensional isotopic fingerprint (Obrist et al., 2018). Atmospheric Hg(0) and Hg(II) in wet

deposition exhibit distinct $\Delta^{199}$Hg and $\Delta^{200}$Hg signatures (Gratz et al., 2010;Chen et al., 2012;Sherman et al., 2012b;Demers et al., 2013;Enrico et al., 2016). Foliar uptake of atmospheric Hg$^{0}$ discriminates heavier Hg isotopes, leading to consistently lower $\delta^{202}$Hg values reported in foliage compared to atmospheric Hg(0) (Demers et al., 2013;Enrico et al., 2016;Obrist et al., 2017;Yu et al., 2017;Yuan et al., 2018). Using the triple isotopic fingerprint to distinguish between Hg(0) and Hg(II) deposition, an increasing number of studies around the globe revealed that 60-90% of Hg in soils is derived from Hg(0) uptake by vegetation

(Demers et al., 2013;Jiskra et al., 2015;Enrico et al., 2016;Zheng et al., 2016;Obrist et al., 2017).

This paper is part of a larger study aiming to better understand the fate of Hg in Arctic tundra ecosystems, centred around a two-year field campaign in interior Alaska. In Obrist et al. (2017), we performed a two-year mass balance of terrestrial-atmosphere exchange over the tundra and showed that Hg(0) uptake by vegetation and soil represents 70% of total atmospheric deposition. We also investigated the spatial distribution of Hg in tundra soils (Olson et al., 2018), and spatial and temporal

patterns in snow (Agnan et al., 2018) and in vegetation (Olson et al., 2019). The goal of the present study is to better understand the processes controlling terrestrial-atmosphere exchange of Hg and the impact of these processes on atmospheric Hg(0) concentrations. For this purpose, we use Hg stable isotopes of Hg(0) in the atmosphere and in interstitial snow and soil pore air to assess the role of Hg(0) uptake by vegetation, Hg(0) oxidation in soils, and re-emission processes. We compare the systematics in Hg stable isotopes during different seasons of the year and study diel variations in combination with Hg(0) flux

measurements and other auxiliary measurements.





## 2 Materials and methods

### 2.1 Study site

The study was conducted at Toolik field station (68° 38′ N, 149° 36′ W) in northern Alaska, USA, 180 km inland from the Arctic Ocean coast. All measurements were conducted on an acidic tussock tundra, on aquiturbels soils with an active layer of

60 – 100 cm (Obrist et al., 2017). The climate of Toolik field station is characterized by low mean annual temperatures of −8.5 °C and mean annual precipitation of 312 mm yr$^{-1}$ (Cherry et al., 2014). During the one-year Hg isotope campaign from October 2015 to September 2016, the tundra was snow-covered during 248 days (Agnan et al., 2018), leading to a relatively short snow-free growing season.

### 2.2 Hg stable isotope sampling and measurement

Hg(0) was continuously sampled in the atmosphere (0.3 m and 2 m above ground), in interstitial air of surface snow (0 m and 0.1 m above ground) and in soil pore air (0.4 m below ground) at low flow rates of 0.2 L min$^{-1}$. Interstitial snow air for Hg(0) stable isotope measurements was sampled from a dedicated snow tower adapted from Seok et al. (2009), consisting of an aluminium construction with three gas inlets on horizontal bars at 0 m, 0.1 m and 0.3 m above ground (Figure S1 A). Two soil wells as described in Obrist et al. (2017) were dedicated to Hg(0) stable isotope sampling of soil air. Each soil well consisted

of two 47mm single stage filter assembly (Savillex, Eden Prairie, USA) with Teflon filter membranes gas inlets, positioned 1.5m apparat in soil pits at 0.4 m depth (Figure S1 B).

Hg(0) was trapped on iodated activated carbon (IAC) traps (Brooks Rand, 0.1 g in custom made 12 cm long glass tubes with an inner diameter of 4mm, (Fu et al., 2014)) and samples were changed manually during site visits every 6-8 weeks. During site visits in March and June 2016, higher temporal resolution (2-4 days) sampling was conducted with higher flow rates of

1.5-2 L min$^{-1}$. Overall total volumes of sampled air per sample ranged from 5.7 to 17.7 m$^3$ (Table S1). During the growing season (June to September 2016), diel variation of atmospheric Hg(0) was assessed using two parallel sampling lines operated with a time switch. Daytime samples were collected from 06:00 to 22:00, and nighttime samples were collected from 22:00 to 06:00. Of the 14 soil pore air samples taken during the one-year Hg isotope campaign, only three samples contained sufficient Hg for isotopic analysis. The reason for this is that, in particular during winter months, the Hg(0) in soil pore air was largely

depleted (below detection limit of ≈0.1 ng m$^{-3}$) (Obrist et al., 2017;Agnan et al., 2018), making isotopic characterization of soil pore air Hg(0) pool impossible during this time period.

The protocol for Hg stable isotope measurements of Hg(0) was adapted from Fu et al. (2014). We used lower amounts of IAC trap material (0.1 g) to reduce possible matrix effects during cold-vapor generation. Breakthrough was tested in the lab and under field conditions by connecting a Tekran 2537 after the IAC trap, and Hg(0) measurements were always below the

detection limit (<0.1 ng m$^{-3}$). IAC traps were combusted in a two-stage oven system, and Hg was recovered in a 40 vol.% 2HNO$_3$:1HCl oxidizing acid trap. Hg stable isotope ratios of trap solutions were measured by cold vapor separation multi-collector inductively coupled plasma mass spectrometry (CV MC-ICP/MS) at the Observatory Midi-Pyrénées, Toulouse (Jiskra et al., 2019;Sun et al., 2013). 6 process blanks and 1 field blank were measured during the sample processing and were on average 0.25 ng Hg/trap (max 0.85 ng Hg/trap), representing 1-2 % of typical Hg amounts collected during sampling

periods. Amounts of Hg collected on IAC-traps sampling atmospheric Hg(0) were compared to Hg(0) concentration measurements with a Tekran 2537 and revealed sample yields of 107 ± 23 % (mean ± 1 SD, n = 22).

Mass dependent fractionation (MDF) of Hg stable isotopes is reported in small delta notation (δ) in per mil (‰) deviation from to the reference NIST 3133 Hg standard:

$$\delta^{xxx}Hg = ({}^{xxx/198}Hg_{sample}/{}^{xxx/198}Hg_{NIST3133} -1) \times 10^3 \tag{1}$$



where 'xxx' refers to measured isotope masses: 199, 200, 201, 202, and 204. Mass independent fractionation (MIF) is reported in capital delta notation ($\Delta$), which is defined as the difference between the measured $\delta^{199}Hg$, $\delta^{200}Hg$, $\delta^{201}Hg$, and $\delta^{204}Hg$ values and those predicted for MDF relative to $\delta^{202}Hg$ using the kinetic MDF law:

$$\Delta^{xxx}Hg = \delta^{xxx}Hg - SF^{xxx} \times \delta^{202}Hg \tag{2}$$

where $SF^{xxx}$ is the mass-dependent scaling factor of 0.252 for $^{199}Hg$, 0.502 for $^{200}Hg$, 0.752 for $^{201}Hg$, and 1.493 for $^{204}Hg$ (Blum and Bergquist, 2007). Hg isotope enrichment factors associated with two pools ($\varepsilon^{xxx}Hg_{pool1-pool2}$ and $E^{xxx}Hg_{pool1-pool2}$) were calculated from the difference in MDF and MIF signature between two pools (pool 1 and pool 2) as follows;

$$\varepsilon^{xxx}Hg_{pool1-pool2} = \delta^{xxx}Hg_{pool1} - \delta^{xxx}Hg_{pool2} \tag{3}$$

$$E^{xxx}Hg_{pool1-pool2} = \Delta^{xxx}Hg_{pool1} - \Delta^{xxx}Hg_{pool2} \tag{4}$$

The MDF enrichment factors of a reaction ($\varepsilon^{xxx}Hg_{reaction}$) was determined by fitting a linear regression model (lm function of R) to the observational data following Mariotti et al. (1981):

$$\delta^{xxx}Hg_{residual} = \delta^{xxx}Hg_0 + \varepsilon^{xxx}Hg_{reaction} \times \ln f \tag{5}$$

where $\delta^{xxx}Hg_{residual}$ corresponds to the Hg isotope signature of the residual Hg(0), $\delta^{xxx}Hg_0$ corresponds to the initial Hg(0)

isotope signature and $f$ to the fraction of Hg(0) remaining in the gas phase. Note that for high $f > 0.4$ the systematic error of this simplified approach is minimal. The MIF enrichment factor ($E^{xxx}Hg_{reaction}$) was calculated as follows:

$$\Delta^{xxx}Hg_{residual} = E^{xxx}Hg_{process} \times \delta^{xxx}Hg_{residual} \tag{6}$$

The long-term precision was assessed through repeated analysis of the ETH-Fluka Hg standard, which yielded values of $-1.44 \pm 0.19$ ‰, $0.08 \pm 0.1$ ‰, $0.02 \pm 0.1$ ‰, $0.02 \pm 0.09$ ‰, $-0.03 \pm 0.2$ ‰ ($2\sigma$, n = 73) for $\delta^{202}Hg$, $\Delta^{199}Hg$, $\Delta^{200}Hg$, $\Delta^{201}Hg$, and

$\Delta^{204}Hg$, respectively, in agreement with published values (Jiskra et al., 2015;Smith et al., 2015). The Almaden standard was measured less frequently and results were -0.58±0.15‰, -0.02±0.09‰, 0.00±0.1‰, -0.06±0.12‰, -0.04±0.23‰ ($2\sigma$, n=21) for $\delta^{202}Hg$, $\Delta^{199}Hg$, $\Delta^{200}Hg$, $\Delta^{201}Hg$, and $\Delta^{204}Hg$, in agreement with previously reported values (Demers et al., 2013;Jiskra et al., 2015;Enrico et al., 2016).

**2.3 Hg(0) flux measurements**

Micrometeorological flux measurements to quantify Hg(0) exchange at the ecosystem level were conducted using an aerodynamic gradient flux method. Surface-atmosphere flux was calculated by measurement of concentration gradients in the atmosphere above the tundra in conjunction with atmospheric turbulence parameters as follows:

$$F_{Hg^0} = -\frac{k \times u_* \times z}{\phi_h\left(\frac{z}{L}\right)} \times \frac{\partial c(Hg(0))}{\partial z} \tag{7}$$

where k denotes the von Karman constant (0.4), $u_*$ the friction velocity, z the measurement height, $\phi_h(z/L)$ the universal

temperature profile, L the Monin-Obukhov length, and $\partial c(Hg(0))/\partial z$ the vertical Hg(0) gas concentration gradient. Hg(0) concentrations at heights of 61 cm and 363 cm above the soil surface were measured through 0.2 μm Teflon® inlet filters connected to ~35 m of perfluoroalkoxy-polymer (PFA) lines. A valve control system with three-way solenoid valves (NResearch, West Caldwell, NJ, USA) allowed switching between the gradient inlets every 10 min. A set of trace gas analysers with a total sampling flow of 1.5 L min$^{-1}$ was connected to the gradient inlets by solenoid valves. The trace analysers included

an air mercury analyser (Model 2537A, Tekran Inc. Toronto, Canada) and a Cavity Ring-Down (CRD) greenhouse gas analyser to measure $CO_2$, $H_2O$, and $CH_4$ (Los Gatos Research, San Jose, USA). Fluxes were calculated only during periods of appropriate turbulence following (Edwards et al., 2005) and as described in (Obrist et al., 2017). For quality control, sampling



line blanks and line inter-comparisons where the two gradient lines were put on the same height were performed approximately every 6-8 weeks (Obrist et al., 2017). The planetary boundary layer (PBL) stability was assessed through the stability index ($\zeta$, dimensionless), where:

$$\zeta = \frac{z}{L} \hspace{8cm} (8)$$

where $z$ represents the height of the sonic above ground and $L$ represents the Monin-Obukhov length. The PBL was considered stable when $\zeta > 0.1$, instable when $\zeta < -0.1$, and neutral for $-0.1 < \zeta < 0.1$ (Peichl et al., 2013). During the sampling period, auxiliary variables showed the following daily average values: air temperature of $-7.4$ °C (from $-40.6$ to $20.4$ °C), relative humidity of 74% (from 37 to 98%), and the wind speed of 2.36 m s$^{-1}$ (from 0 to 7.82 m s$^{-1}$).

### 3 Results and Discussion

We divide the presentation and discussion of results into three seasons of the year. In winter (20 Oct 2015 to 17 Mar 2016), the tundra site was continuously snow-covered and the climatic conditions were characterized by low temperatures (mean = $-17.8$ °C, hourly max = 1.2 °C) and low solar radiation (mean = 0.02 kW m$^{-2}$, hourly max = 0.41 kW m$^{-2}$). In spring (17 Mar 2016 – 05 Apr 2016), temperatures were low (mean = $-19.2$ °C, hourly max = $-2.4$ °C) and the tundra was still snow-covered, however solar radiation increased (mean = 0.13 kW m$^{-2}$, max = 0.54 kW m$^{-2}$) and occasional atmospheric mercury depletion

events (AMDE) were detected at the study site (Obrist et al., 2017;Agnan et al., 2018). During summer (03 May 2016 – 09 Sep 2016), air temperature was above freezing (mean = 6.7 °C, hourly max = 25.1 °C), solar radiation was high (mean = 0.19 kW m$^{-2}$, hourly max = 0.80 kW m$^{-2}$), and the study site was predominantly free of snow.

### 3.1 Wintertime

Over the winter period, atmospheric Hg(0) concentrations and $CO_2$ mixing ratios were relatively constant and there was little

diel variation (Figure 1 A, Figure S2). Low solar radiation led to relatively stable PBL conditions throughout the day (Figure 1 B). Hg(0) flux measurements revealed a small deposition (mean: $-0.26$ ng m$^{-2}$ h$^{-1}$) (Obrist et al., 2017). The Hg(0) net deposition flux is supported by observed depletions of atmospheric Hg(0) in interstitial snow air (0.69 ± 0.22 ng m$^{-3}$, mean ± 1SD, 0 and 0.1 m sampling height) relative to atmospheric levels (1.07 ± 0.04 ng m$^{-3}$, mean ± 1SD) (p = 0.02, two-sided t-test), implying a net Hg(0) sink in the ecosystem (Figure 2 C). The depletion of atmospheric Hg(0) in interstitial snow air was

associated with an increase in δ$^{202}$Hg (1.08‰ ± 0.20 ‰ versus 0.77 ‰ ± 0.16 ‰ in ambient Hg(0), mean ± 1SD, p = 0.02, two-sided t-test) (Figure 2 A) and a decrease in Δ$^{199}$Hg ($-0.31$ ‰ ± 0.05 ‰ versus $-0.23$ ‰ ± 0.06 ‰ in ambient Hg(0), mean ± 1SD, p = 0.04, two-sided t-test) (Figure 2 B). Hg(0) dry deposition to surface snow (Douglas and Blum, 2019) and by vegetation uptake (Demers et al., 2013;Enrico et al., 2016;Obrist et al., 2017) has been reported to discriminate heavier Hg(0) isotopes consistent with these observations in the interstitial snow air. A wintertime Hg(0) sink can either occur by Artic snow,

soil, or vegetation still active under snowpack. From a mass balance perspective, Hg(0) dry deposition to snow is considered to only play a minor role in the interior arctic tundra. For example, using snow data in Agnan et al. (2018), we calculated a total seasonal snow Hg pool of only 50 ng m$^{-2}$ at Toolik Field station. Assuming that all this Hg in the snow was originating from the dry deposition of Hg(0), this would account for <10% of the Hg(0) deposition during the snow-covered period (total of 2.4 µg m$^{-1}$ yr$^{-1}$ (Obrist et al., 2017)). Recently, Douglas and Blum (2019), however, suggested that Hg(0) dry deposition to

snow was the major source of Hg in meltwater collected on the coast of the Arctic Ocean close to Barrow, ~400 km north-west of Toolik field station. In snow on the coast of the Arctic Ocean, concentrations of halogens, which are considered to mediate reactive Hg(0) uptake, are elevated compared to inland sites (Douglas and Sturm, 2004;Agnan et al., 2018;Douglas et al., 2017) leading to much higher snow Hg pools (~2000 ng m$^2$) in coastal snowpack (Douglas et al., 2017).





Both MDF and MIF signatures in interstitial snow air Hg(0) are complementary (i.e., in opposite direction) to the Hg isotope signatures observed in lichen at the same site ($\delta^{202}Hg = -0.80$ ‰ $\pm 0.20$ ‰, $\Delta^{199}Hg = 0.20$ ‰ $\pm 0.21$‰, mean $\pm$ 1SD, n = 12) (Olson et al., 2019). Lichen can actively exchange $CO_2$ for photosynthesis under snow cover (Kappen, 1993), and could possibly take up atmospheric Hg(0) also during winter months. In contrast, Hg(0) oxidation by humic acids in soils would lead

to more negative $\delta^{202}Hg$ values (Zheng et al., 2018) (see discussion below in section 3.3.3), and hence would be inconsistent with the observed enrichment in heavier Hg(0) isotopes in the interstitial snow air (Figure 2 A and Figure 8).

### 3.2. AMDE season

During spring 2016, three major AMDE's occurred at Toolik Field station (Figure 3). During the first AMDE (19 Mar 2016 - 20 Mar 2016, event 1 in Figure 3), atmospheric Hg(0) concentrations dropped below detection limit (<0.1 ng m$^{-3}$), while

atmospheric Hg(II) concentrations remained relatively low with the exception of individual spikes up to 0.4 ng m$^{-3}$. During the second AMDE (26 Mar 2016 – 29 Mar 2016, event 2 in Figure 3) atmospheric Hg(0) concentrations temporally decreased to 0.75 ng m$^{-3}$ and Hg(II) concentrations remained enhanced around 0.2 ng m$^{-3}$ for two days while turbulent PBL conditions prevailed. During the third AMDE (1 Apr 2016 – 3 Apr 2016, event 3 in Figure 3), atmospheric Hg(0) concentrations at times decreased below detection limit (<0.1 ng m$^{-3}$) while Hg(II) concentrations remained high around 0.4 ng m$^{-3}$ for two days.

During events 1 and 3, $O_3$ mixing ratios dropped below 10 ppb, whereas during event 2, $O_3$ remained high (>30 ppb). Van Dam et al. (2013) reported that during AMDEs and ODEs observed at Toolik field station, similar AMDEs events were also observed on the coast in Barrow ~400 km to the north-west. AMDEs and ODEs are driven by bromine emissions from the Arctic ocean and transported to different extents to the interior tundra (Van Dam et al., 2013). For example, during event 1 no elevated Hg(II) concentrations were observed, thus we assume that air depleted in Hg(0) was transported to Toolik field station

while the deposition of Hg(II) likely occurred closer to the coast.

Figure 3D displays measured Hg(0) fluxes during these AMDE periods (17 Mar 2016 – 5 Apr 2016) showing strong Hg(0) re-emission after the three AMDEs. On average, a net Hg(0) re-emission of 1.5 ng m$^{-2}$ h$^{-1}$ was measured, making the time of AMDEs the only period of the year where net Hg(0) re-emission occurred (Obrist et al., 2017). Strong Hg(0) re-emission from the snowpack has been reported during and after AMDEs due to fast reduction of Hg(II) deposition (Johnson et al.,

2008;Douglas et al., 2012).

Snowmelt occurred in May in 2016 and snow height quickly declined between May 7 (24 cm) and May 13 (0 cm). In contrast to wintertime patterns, the interstitial snow air Hg(0) during snowmelt showed low $\Delta^{199}Hg$ values of −0.62 ‰ and −0.44 ‰ versus −0.23 ‰ $\pm$ 0.06 ‰ in ambient Hg(0) (Figure S3B). The negative $\Delta^{199}Hg$ values in Hg(0) suggest a substantial contribution of Hg(0) re-emission after photoreduction of Hg(II) in snow, which exhibited negative $\Delta^{199}Hg$ values with a

minimum of −1.37‰ (Obrist et al., 2017). This observation is consistent with chamber experiments, where negative $\Delta^{199}Hg$ of −2.08 ‰ were reported for Hg(0) re-emission from snow (Sherman et al., 2010).

Even-MIF ($\Delta^{200}Hg$) is considered not to be affected by post-deposition processes such as re-emission (Sherman et al., 2010;Enrico et al., 2016), providing a conservative tracer for the pathway of atmospheric Hg deposition. $\Delta^{200}Hg$ values measured in snow impacted by AMDEs at Toolik field station and other sites in Alaska (Obrist et al., 2017;Sherman et al.,

2010;Sherman et al., 2012a) are similar to the $\Delta^{200}Hg$ values of atmospheric Hg(0) (Figure 6) (−0.06‰ $\pm$ 0.06‰ versus −0.05‰ $\pm$ 0.04‰, mean $\pm$ 1SD). This similarity can be explained by a quantitative oxidation of atmospheric Hg(0) to Hg(II) (e.g., event 1 in Figure 3) that is deposited to snow. Hg(II) in snow thereby inherits the isotopic composition of the source Hg(0) due to conservation of mass, irrespective of the isotopic fractionation factor associated with Hg(0) oxidation. Several samples exhibited $\Delta^{200}Hg$ values between that of atmospheric Hg(0) and Hg(II) in precipitation measured in temperate regions

(Figure 4). This intermediate $\Delta^{200}Hg$ signature can be explained by either AMDEs with non-quantitative oxidation due to limited Br oxidant availability (e.g., event 2 in Figure 3), or a mixing of AMDE Hg(II) with non-AMDE Hg(II) present in the





overlying Arctic free troposphere. Even MIF ($\Delta^{200}Hg$) has been suggested as a promising tracer to distinguish between atmospheric deposition of Hg(II) in precipitation, which exhibits positive $\Delta^{200}Hg$ anomalies, and direct Hg(0) deposition (e.g., uptake by vegetation), which exhibits small negative $\Delta^{200}Hg$ (Enrico et al., 2016;Sun et al., 2019). We caution that the presence of AMDEs complicates the use of $\Delta^{200}Hg$ for mixing model based Hg deposition calculations in the Arctic (Obrist et al., 2017).

### 3.3 Summertime

#### 3.3.1 Drivers of diel cycling in atmospheric Hg(0)

Figure 5 represents a time series of atmospheric Hg(0) concentration and $CO_2$ mixing ratio, PBL stability and Hg(0) fluxes during mid-summer (15 Jul 2016 – 28 Jul 2016). Atmospheric Hg(0) concentrations generally declined during each night, and strongest Hg(0) depletions (Hg(0) <1 ng m$^{-3}$) were observed when the PBL was stable ($\zeta > 0.1$, events 2 and 3 and green bars

in Figure 5 B). These Hg(0) nighttime minima coincided with maxima of atmospheric $CO_2$ mixing ratios of 410 to 420 ppm. These patterns are consistent with measured Hg(0) deposition fluxes during nights (daily minima at 03:00, $-2.56 \pm 0.35$ ng m$^{-2}$ h$^{-1}$ (mean $\pm$ 1SD), Figure 1I), and $CO_2$ accumulation in the PBL driven by nighttime $CO_2$ soil respiration (Wofsy et al., 1993;Schlesinger and Andrews, 2000;Grant and Omonode, 2018). During nights with unstable PBL conditions (e.g., event 1 in Figure 5), diel Hg(0) and $CO_2$ variations were much lower or absent due to increased mixing with free tropospheric air

containing background levels of Hg(0) and $CO_2$. During late summer (Figure S4, 20 Aug 2016 – 31 Aug 2016), the longer duration of stable nocturnal PBL conditions led to even more pronounced nighttime depletions in Hg(0). During daytime under strong solar radiation, flux measurements showed net Hg(0) emission around noon (daily maxima at 12:00, $3.2 \pm 0.83$ ng m$^{-2}$ h$^{-1}$ (mean $\pm$ 1SD), Figure 1I). Daytime Hg(0) re-emission, however, did not lead to a build-up of atmospheric Hg(0) above the surface due to prevailing turbulent conditions allowing efficient mixing with background free tropospheric air. These patterns

demonstrate how atmospheric Hg(0) and $CO_2$ are both controlled by the magnitude and direction of the net ecosystem exchange fluxes in conjunction with PBL stability. Overall, flux measurements showed the tundra ecosystem to be a net sink of atmospheric Hg(0) over the duration of the growing season (mean: $-0.12$ ng m$^{-2}$ h$^{-1}$).

#### 3.3.2 Hg isotope fractionation during foliar uptake of atmospheric Hg(0)

Figure 6 shows a scatterplot of atmospheric Hg(0) concentrations and the $\delta^{202}Hg$ values sampled during different times of the

day in summer 2016. Hg(0) sampled during the night was characterized by a lower Hg(0) concentration ($1.06 \pm 0.13$ ng m$^{-3}$), a higher $\delta^{202}Hg$ ($1.31$ ‰ $\pm 0.15$‰), and a similar $\Delta^{199}Hg$ ($-0.28$ ‰ $\pm 0.08$ ‰) (22:00 – 06:00, mean $\pm$ 1SD, n = 4), compared to Hg(0) sampled during the day, with Hg(0) = $1.16 \pm 0.11$ ng m$^{-3}$, $\delta^{202}Hg = 1.07$ ‰ $\pm 0.19$‰ , $\Delta^{199}Hg = -0.24$ ‰ $\pm 0.08$‰ (06:00 - 22:00, mean $\pm$ 1SD, n = 2). An enrichment of heavy atmospheric Hg(0) isotopes during nights is consistent with the preferential uptake of light Hg(0) isotopes by vegetation (Demers et al., 2013;Enrico et al., 2016;Obrist et al., 2017;Olson

et al., 2019;Yuan et al., 2018;Yu et al., 2016), the dominant Hg(0) deposition pathway at the study site (Obrist et al., 2017). The enrichment factor $\varepsilon^{202}Hg$ for vegetation uptake of Hg(0), determined based on Figure 6, was $-4.22$ ‰ $\pm 1.01$ ‰ (mean $\pm$ 1 se, R$^2$ = 0.68, p = 0.003) (Figure S5). Enrico et al.(2016) estimated a $\varepsilon^{202}Hg_{plant-air}$ of $-2.6$ ‰ for foliar uptake by sphagnum moss, using a Rayleigh model to fit the atmospheric Hg(0) concentration and $\delta^{202}Hg$ measured at two locations, a mountain site unaffected by local terrestrial-atmosphere exchange and a peat bog where Hg(0) in air was depleted by foliar uptake.

Similarly, observations of the difference in $\delta^{202}Hg$ between plants and atmospheric Hg(0) suggested $\varepsilon^{202}Hg_{plant-air}$ between $-1$ and $-3$‰ (Demers et al., 2013;Enrico et al., 2016;Obrist et al., 2017;Olson et al., 2019;Yuan et al., 2018;Yu et al., 2016). At Toolik field station, the difference between $\delta^{202}Hg$ in vegetation relative to atmospheric Hg(0) was also considerably lower (range of $-1.29$ to $-2.09$ ‰, depending on vegetation species (Olson et al., 2019)), than the fractionation factor derived from the atmospheric pattern.





This discrepancy can be explained by the fact that $\delta^{202}$Hg signatures measured in vegetation do not only reflect the isotopic fractionation during foliar uptake but also contain a re-emission component. Hg(II) reduction is expected to lead to more positive $\delta^{202}$Hg values in the residual, foliar Hg(II) fraction, irrespective of the reduction mechanism (Bergquist and Blum, 2007;Zheng and Hintelmann, 2010;Kritee et al., 2007;Jiskra et al., 2015). Re-emission of foliar Hg is supported by

observed negative shifts in odd-mass isotope MIF ($\Delta^{199}$Hg) in vegetation relative to $\Delta^{199}$Hg of atmospheric Hg(0) which have been observed at Toolik field station (Olson et al., 2019) and elsewhere (Enrico et al., 2016;Demers et al., 2013). Positive $\Delta^{199}$Hg in atmospheric Hg(0) re-emitted from foliage has recently been constrained by flux bag experiments (Yuan et al., 2018). It is therefore expected that the fractionation factor of foliar uptake is larger than just the difference between $\delta^{202}$Hg in foliage and Hg(0) in the atmosphere. Instead, the difference reflects a net fractionation consisting of the isotopic fractionation

during foliar uptake, as well as during foliar reduction and re-emission.

Our observation that the fractionation factor derived from $\delta^{202}$Hg of atmospheric Hg(0) (Figure 6) is larger than the difference of $\delta^{202}$Hg between vegetation and the atmosphere could also be associated with the diel variation of Hg(0) fluxes and PBL dynamics. During daytime, atmospheric turbulence is higher and therefore local signals of terrestrial re-emission are expected to be diluted by mixing with background Hg(0). At night, when the PBL is stable, foliar uptake of lighter Hg(0) isotopes is

imprinted on the residual atmospheric Hg(0).

### 3.3.3 Sink of Hg(0) in soil inferred from Hg stable isotopes

At our study site, soil pore air Hg(0) concentrations were below ambient levels measured in the atmosphere all year (Obrist et al., 2017), with an average concentration of $0.54 \pm 0.14$ ng m$^{-3}$, indicating a consistent sink of Hg(0) in soils. Hg(0) in soil pore air showed a lower $\delta^{202}$Hg ($-0.01$ ‰ $\pm 0.39$ ‰) and a higher $\Delta^{199}$Hg ($-0.18$ ‰ $\pm 0.07$ ‰) (mean $\pm$ 1SD, n = 3)

compared to ambient atmospheric Hg(0) (Hg(0) = $1.1 \pm 0.09$ ng m$^{-3}$, $\delta^{202}$Hg = 0.81 ‰ $\pm 0.18$ ‰, $\Delta^{199}$Hg = $-0.25$ ‰ $\pm$ 0.04‰; 24 h, mean $\pm$ 1SD, n = 6) during the summer and fall period when we were able to quantify soil pore air isotope patterns. Fitting the Hg stable isotope fractionation trajectory for MDF and MIF of three data points of soil pore air samples and the atmospheric Hg(0) samples resulted in enrichment factors of $\varepsilon^{202}$Hg$_{\text{soilair-atmosphere}}$ = $-1.00$ ‰ $\pm 0.25$‰ (mean $\pm$ 1 se, $R^2$ = 0.69, p=0.005) (Figure 7 A) and E$^{199}$Hg$_{\text{soilair-atmosphere}}$ = 0.07 ‰ $\pm 0.04$ (mean $\pm$ 1 se, $R^2$ = 0.32, p = 0.11) (Fig. 7 B).

Recently, (Zheng et al., 2018) investigated Hg stable isotope fractionation during oxidation of dissolved Hg(0) by low molecular weight thiol compounds and natural humic acids (HA). For oxidation by HA ,they reported an enrichment of light Hg(0) isotopes ($\varepsilon^{202}$Hg$_{\text{Hg(0)-Hg(II)}}$ = $-1.54$ ‰ $\pm 0.05$ ‰, mean $\pm$ 1 se) and a positive odd-mass Hg MIF (E$^{199}$Hg$_{\text{Hg(0)-Hg(II)}}$ = -0.18 ‰ $\pm$ 0.03 ‰, mean $\pm$ 1 se) in the residual Hg(0) fraction (Zheng et al., 2018). Our limited number of soil air measurements (n = 3) are in agreement with the fractionation trajectory for HA oxidation (red straight lines in Figure 7), suggesting abiotic

oxidation in soils as a cause for observed soil Hg(0) uptake. In spite of a consistent soil Hg(0) sink in soils, Obrist et al. (2014) estimated that the soil Hg(0) sink results in only minor Hg(0) soil uptake (<-0.03 ng m$^{-2}$ h$^{-1}$ ) due to low diffusivity. We estimate that at Toolik field station, such a soil Hg(0) sink would only account for <5 % of the total Hg(0) deposition flux observed. This is consistent with Hg stable isotope data showing an opposite direction of soil air $\delta^{202}$Hg compared to interstitial snow air Hg(0) in respect to wintertime atmospheric Hg(0) (Figure 8), suggesting that minor soil uptake of Hg(0) does not

significantly modify interstitial Hg(0) patterns in the snowpack above.

### 4 Conclusions

We document that atmospheric Hg(0) concentrations and the isotopic composition are strongly affected by terrestrial-atmosphere exchange of Hg(0) in an Arctic tundra ecosystem. While directions and magnitudes of the terrestrial-atmosphere exchange varies with season and time of the day, atmospheric stability and the dynamics of the PBL strongly affects



atmospheric Hg(0) concentrations. In Arctic winter, in the absence of light and with permanent snow cover, the terrestrial-atmosphere exchange of the tundra ecosystem shows a small but steady net Hg(0) deposition. Atmospheric Hg(0) concentrations were relatively constant throughout the day as a consequence of stable PBL conditions and little variability in surface exchange fluxes. During several weeks in spring when AMDEs were present, terrestrial-atmosphere exchange was

dominated by photochemically-driven Hg(0) re-emission from the snow surface, leading to higher atmospheric Hg(0) concentrations during the daytime when solar radiation was higest. During the growing season, the terrestrial-atmosphere exchange was dominated by vegetation uptake of Hg(0). Hg(0) re-emission from vegetation and possibly from soil surfaces counteracted Hg(0) uptake by vegetation, leading to strong diel Hg(0) variations with net Hg(0) deposition during the night and early morning and net Hg(0) emission around noon during high solar radiation. Hg(0) concentrations were continuously

depleted in interstitial snow and soil pore air. The first Hg stable isotope measurements of Hg(0) in soil air indicated that the soil Hg(0) sink was driven by Hg(0) oxidation by natural organic matter. However, we observed no isotopic traces of this Hg(0) soil sink in the interstitial snow air and atmosphere above. Atmospheric Hg(0) isotope systematics were dominated by vegetation uptake of Hg(0) discriminating heavy Hg(0) isotopes in the residual atmospheric pool, which manifested itself most strongly during the vegetation growth season in summer and during stable PBL conditions at night. We found a larger

discrimination of heavier Hg(0) isotopes during foliar uptake when deriving a  fractionation factor from atmospheric Hg(0) observations, compared to deriving this factor based on the difference measured between bulk Hg in vegetation and atmospheric Hg(0). While this discrepancy is not fully understood, it may be attributed to photoreduction and re-emission of lighter Hg(0) isotopes during the day. Overall the study shows the potential of using Hg stable isotopes to better understand the mechanisms driving terrestrial-atmosphere exchange of mercury.

**Acknowledgments**

This project was funded by H2020 Marie Sklodowska-Curie grant 657195 and Swiss National Science Foundation grant PZ00P2_174101 to MJ, European Research Council grant ERC-2010-StG_20091028 to JES and National Science Foundation, award numbers 1304305, 1739567, and 1848212 to DO.

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



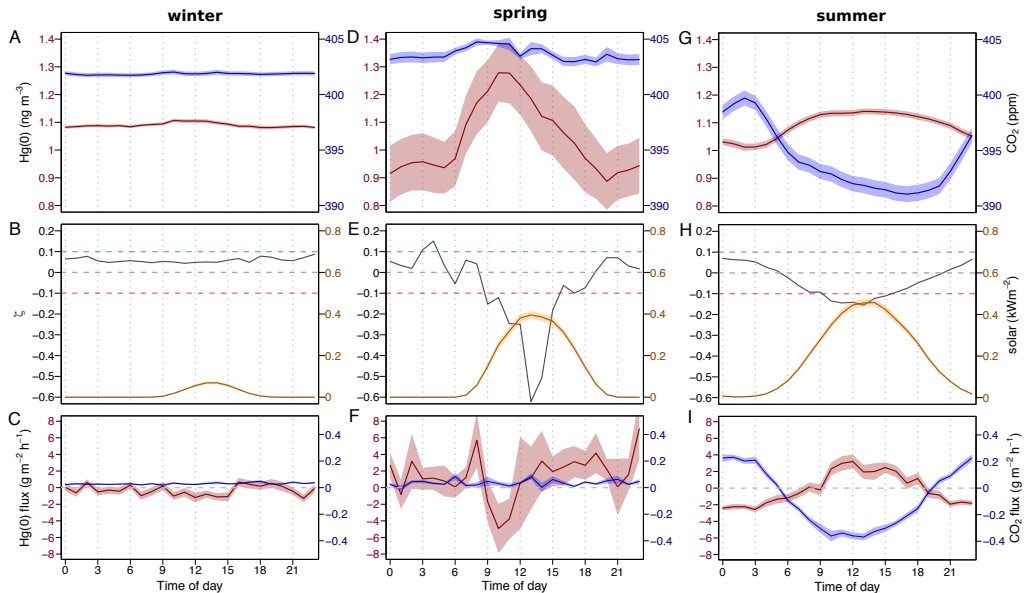

**Figure 1: Diel variation during different seasons of the year of A, D and G: average Hg(0) concentration in red and CO₂ mixing ratio in blue in the atmosphere (2 m height). B, E, and H: median planetary boundary layer stability parameter (ζ, grey), where positive values represent stable conditions, and solar radiation (yellow), C, F, I: average Hg(0) flux in red and CO₂ flux in blue. The shaded areas in A and C represent the mean ± 1 SD interval of concentration and flux measurements, respectively.**

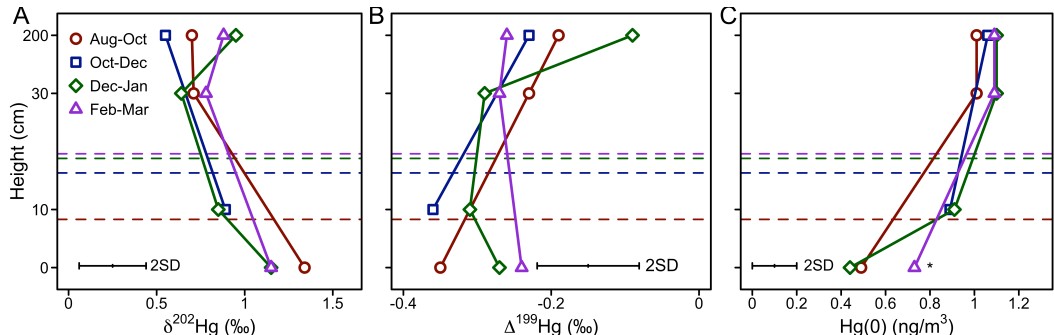

**Figure 2: Hg(0) measurements in interstitial snow air profiles A: mass dependent fractionation stable isotope signature of Hg(0) (δ²⁰²Hg), B: mass-independent fractionation stable isotope signature of Hg(0) (Δ¹⁹⁹Hg), and C: mean Hg(0) concentration. Note that the Hg(0) concentration marked with * was calculated from Hg recovered on the IC-traps, while other Hg(0) concentration profiles were measured by an automated trace gas system deployed in the snowpack (Agnan et al., 2018). The dashed horizontal lines represent the average snow height during the respective period.**



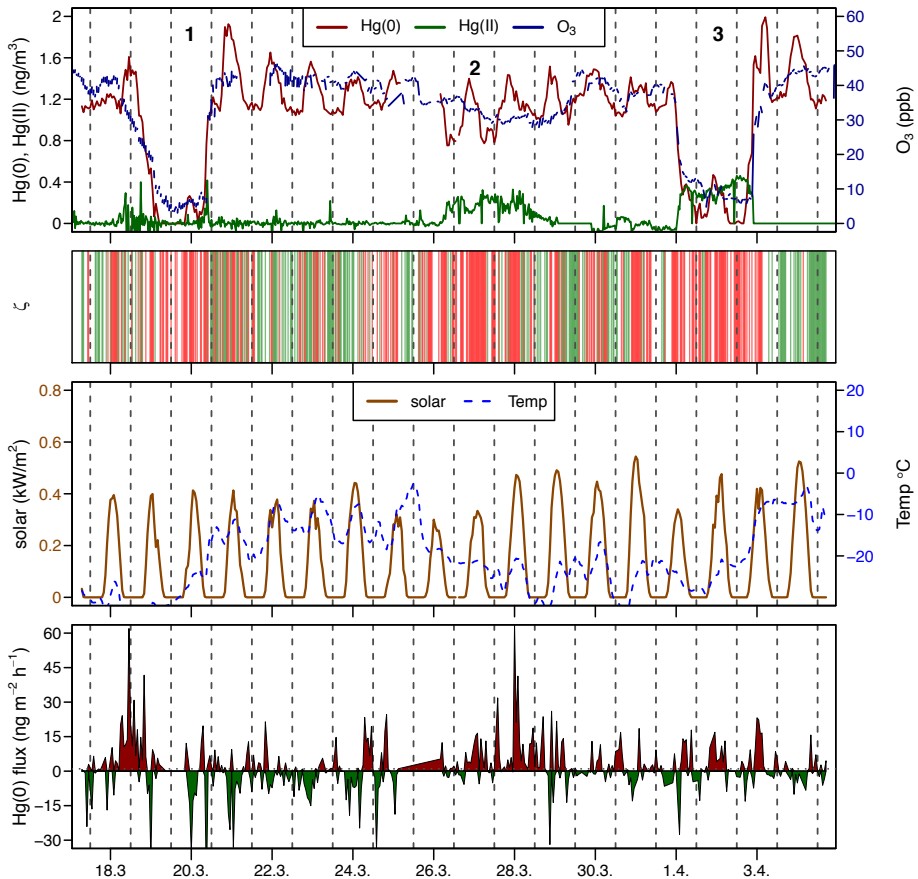

**Figure 3: Time series during springtime AMDEs period (18.3.2016-4.4.2016), A: atmospheric Hg(0) (red), atmospheric Hg(II) (green) concentration, and O₃ mixing ratio (blue), B: planetary boundary layer stability (ζ), where the shaded areas in green represents stable conditions (ζ > 0.1) and shaded areas in red represent turbulent conditions (ζ < -0.1), C: solar radiation and air temperature, and D: Hg(0) flux, where Hg(0) deposition is in green and Hg(0) re-emission is in red. Midnight is indicated by dashed lines. Strong AMDEs when Hg(0) concentrations dropped (1,3) or Hg(II) concentrations increased (2) are marked by numbers.**



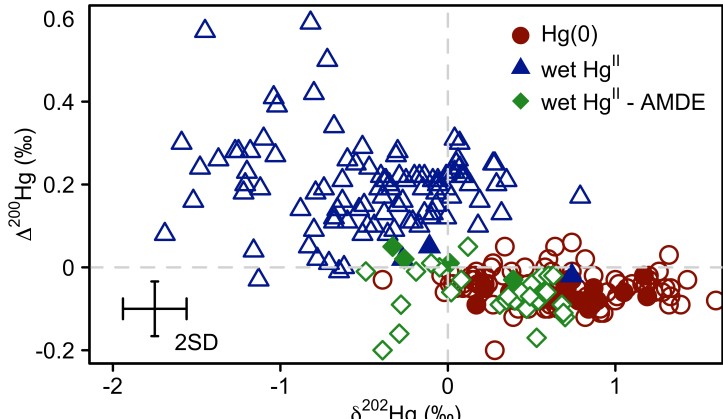

**Figure 4: Mass dependent fractionation (δ²⁰²Hg) versus even mass-independent fractionation (Δ²⁰⁰Hg) of atmospheric Hg(0) (red circles) Hg(II) in wet deposition sampled at locations or during seasons when no AMDEs occurred (blue triangles), and Hg(II) in Arctic show sampled during springtime AMDE season (green diamonds) for the Arctic tundra at Toolik field station (filled symbols), and reported from elsewhere (empty symbols). Atmospheric Hg(0) data are from (Gratz et al., 2010;Demers et al., 2013;Demers et al., 2015;Fu et al., 2016a;Enrico et al., 2016;Obrist et al., 2017), Hg(II) in wet deposition are from (Demers et al., 2013;Sherman et al., 2015;Chen et al., 2012;Gratz et al., 2010), and Hg(II) in snow from AMDEs are from (Obrist et al., 2017;Sherman et al., 2010;Sherman et al., 2012a). Error bars represent the analytical precision determined by the 2 SD from multiple measurements of an inhouse standard.**





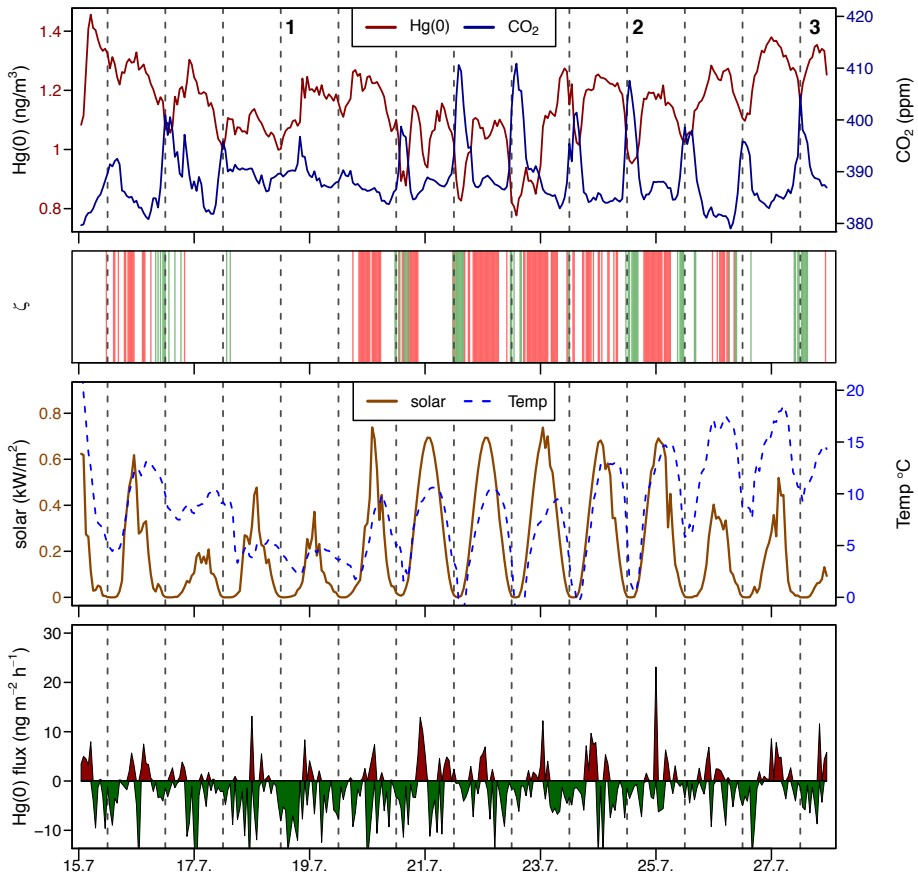

**Figure 5:** Time series during a midsummer period (15.7.2016 – 28.7.2016), A: atmospheric Hg(0) concentration and $CO_2$ mixing ratio, B: median planetary boundary layer stability ($\zeta$), where the shaded areas in green represents stable conditions ($\zeta > 0.1$) and shaded areas in red represent turbulent conditions ($\zeta < -0.1$), C: solar radiation and air temperature, and D: Hg(0) flux, with Hg(0) deposition in green and Hg(0) re-emission in red. Midnight is indicated by dashed lines.

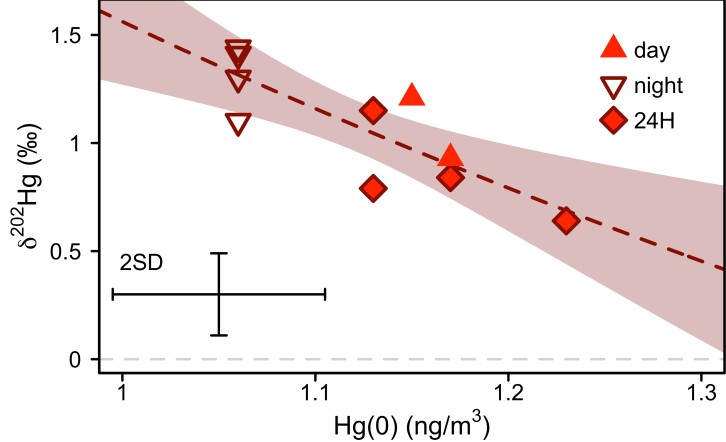

**Figure 6:** Mass dependent Hg isotope signature ($\delta^{202}$Hg) of atmospheric Hg(0) versus Hg(0) concentration during the snow-free growing period (11 Jun 2016 – 10 Sep 2016). The dashed line represents a non-linear Rayleigh fit and the shaded area the 95% confidence interval (see main text). Error bars represent the analytical precision determined by the 2 SD from multiple measurements of an in-house standard.

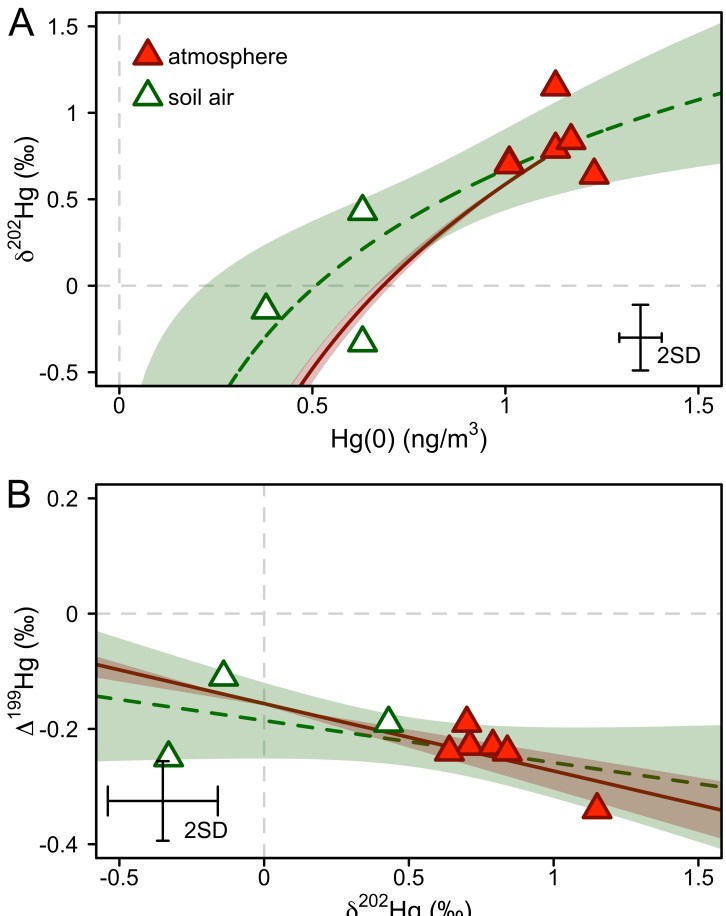

**Figure 7: Hg stable isotope systematics of Hg(0) in the atmosphere during summer/fall and in soil pore air measured during the same time period A: Mass dependent Hg isotope signature ($\delta^{202}$Hg) against Hg(0) concentration B: Mass independent Hg isotope**
5 **signature ($\Delta^{199}$Hg) against mass dependent Hg isotope signature ($\delta^{202}$Hg). The dashed green lines represent the regression of a Rayleigh model (A) and a linear model (B). The straight red lines represent the trajectories for abiotic dark oxidation of Hg(0) by natural humic acids from Zheng et al. (2018). For comparison with observations, the intercept of the linear regressions was adjusted to fit through the average of atmospheric Hg(0). The shaded areas represent the 95% confidence interval. Error bars represent the analytical precision determined by the 2 SD from multiple measurements of an in-house standard.**





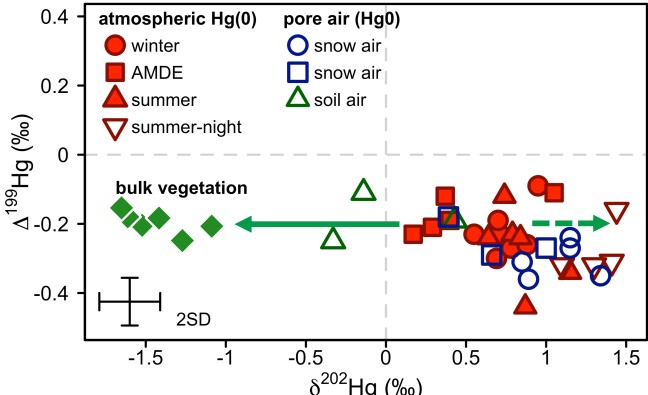

**Figure 8: Overview of Mass independent Hg isotope signature ($\Delta^{199}$Hg) against mass dependent Hg isotope signature ($\delta^{202}$Hg) of atmospheric Hg(0) (red) and Hg(0) in pore air of snow (blue) and soil (green) during winter (circles), AMDE season (squares) and summer/fall (triangles). Bulk vegetation measurements (green diamonds) are reproduced from Obrist et al. (2017). The straight arrow represents the fractionation during Hg(0) uptake by vegetation, the dashed arrow represents the expected development of the corresponding residual Hg(0) pool. Error bars represent the analytical precision determined by the 2 SD from multiple measurements of an in-house standard.**