# Peer review of "Insights from mercury stable isotopes on terrestrial – atmosphere exchange of Hg(0) in the Arctic tundra"

_Biogeosciences, 2019_

## Referee Comment (RC1) · Anonymous Referee #1 · 29 Jun 2019

The reviewer's comments are appended as a PDF-file.

[Figure]

The manuscript is well written, exemplarily concise and of high scientific quality. One problem is, however, that the data presented already to some degree been published in Obrist et al. (2017) doi: 10.1038/nature22997.

The present manuscript refers to this Nature paper more than 20 times, which hampering a throughout reading obtaining comprehensive information from text, tables and figures.

A basic issue is that a summary tabulation of flux and ancillary data statistics (number of observations, flux data coverage (%), % of data rejected due undeveloped turbulence or fetch limitations etc. etc.) is missing in both papers. Please, provide a table in the main part or in a supplement.

The uncertainty in flux measurements is not mentioned and quantified. Such a discussion should also include that the flux derivation is obtained by asynchronous Hg0 sampling of the two heights.

The measured Hg0 deposition velocities should be mentioned and discussed with literature data.

Correlation analysis between measured gases, flux and environmental parameters is not presented.

To improve the readability, consider assigning the oxidation state of Hg in delta and capital delta notations (e.g. $\delta^{202}Hg^0$, $\Delta^{199}Hg^{II}$) when found appropriate.

**Specific comments:**

| | |
|---|---|
| Page 2, Line 1 | *drawn down*, consider revising |
| Page 2, Line 14 - 15 | Lindberg *et al.* 1998 is outdated (suggesting foliage as net source of $Hg^0$). Consider e.g. Bash and Miller (AE, 2009) or Castro et al. (Atmosphere, 2016) |
| Page 3, Line 16 | "1.5m apparat", mistake? |
| Page 4, Line 25 | an aerodynamic… consider the aerodynamic… |
| Page 4, Line 29 - 30 | $\Phi h$ the universal temperature profile, provide a reference for the mathematical form used. |
| Page 5, Line 21 | Provide ±SD of the mean |
| Page 6, Line 10 | … remained relatively low… try to be more concise (numbers) |
| Page 6, Line 16 | ODE's without explanation. Define Ozone depletion events as ODEs. |
| Page 6, Line 22 | Provide median also, if there is a substantial difference with mean |

**Fig. 1.**

---

## Referee Comment (RC2) · Anonymous Referee #2 · 16 Jul 2019

This manuscript is well written and represents the work of a strong field and laboratory team focused on mercury deposition to the Arctic. The study is well conceived and presented. It will be of interest to a variety of leadership, particularly because there is an increasing interest in understanding the source and ultimate fate of Hg in permafrost soils. I have some small to moderate comments/recommendations:

Title: "Insights" is not a strong word for this study. I recommend a far better title. "Mercury stable isotopes reveal XYZ on the terrestrial-atmosphere exchange of Hg(0) in arctic tundra"

[Figure]

This brings me to a question about the conclusions (more later): what does this study say about the seasonal net in versus out of Hg with respect to the snow pack, inferring snow melt, and summer soils? I kept hoping they would provide a seasonal diagram with the Hg % deposited, % re-emitted, and overall fluxes for their site.

Page 1: 12: in arctic mercury Make sure "Arctic" versus "arctic" are correct You could say "the Arctic mercury cycle" 17-18 net emission fluxes based on the AMDEs or over the entire spring there was an overall net loss? 32: Hg emission 33: such as the Arctic, through 36: (AMDEs), leads

Page 2: 32: Toolik Lake is on the Arctic Coastal Plain of Alaska. Not the Interior.

Page 3: 15: an assembly with two 47mm diameter single stage filters (?) membrane 16: apart 18: Since the site visits were every 6-8 weeks: did the filter void spaces fill up? 20: The 5.7 to 17.7 m3. Is this for the long term or short term deployments? 22: no comma after 22:00 24: what is "sufficient Hg for analysis"? 26: the soil pore 29: no comma after IAC trap 30: no comma after oven system

Page 4: 11: were determined 31: Here Teflon has the registered trademark but not earlier when "Teflon" is written (page 3, line 15)

Page 5: 5: do you mean sonic sounder? There is little information on the meteorological measurement instrumentation except perhaps here?

Was wind direction measured and analyzed? Any association between MDEs and prior wind from the coast? Any association at all with wind direction and the Hg values measured?

Results and Discussion Were there any measurements during snow melt? How/why were the different time intervals selected? They seem arbitrary. Perhaps non-AMDE winter and AMDE conditions instead of winter and spring? The "spring" is actually colder than "winter."

Where are the data from 5 April to 3 May?

Was snow melt part of the 3 May to 9 Sep timeframe? Or an inundated tundra surface following melt?

Any relationships between summer seasonal thaw and Hg?

30: under the snowpack

38: coastal snowpacks

Page 6: 4: remove "also" as "possibly" is already in there 8 (Figure 3): How were AMDEs defined?

The main text Figure order is 1, 2, 8, 3, 6, 4, 5, 7 Please reorder in numerical order

16: similar AMDE events

8-20: Was there any analysis of the wind back trajectories or the Barrow (now Utqi-agÌG̃vik) based GMD ozone to identify whether the AMDEs were regional to the coast?

Page 7: 8-9: each night, and the strongest

24-28: Here and elsewhere where these types of data are presented. Are the different pools statistically significantly different? Providing the analytical errors is helpful but a statistical analysis of these data is in order. From a visual perspective the standard deviations likely cross over and there is no significant difference.

Page 8 24: data in Figure 7. Same comment as above about statistical analyses

39: strongly affect

Conclusions: I really like the information in this manuscript and how it is presented. There is a lot of work here.

However, the conclusions read like a summary of the results. This study could go far in identifying the seasonal aspects of Hg deposition and re-emission but the authors mostly just summarize. All the way back to the title word "insights" I recommend they go farther.

What can they say about the Hg seasonality of deposition in the Arctic?

In the introduction the authors start with Hg being a pollutant and then introduce AMDEs and talk about snowpack re-emission. A large question there is- what fraction of snowpack Hg makes it into runoff and of that how much ends up stored in soils?

They then mention tundra soils can draw down summertime Hg(0). So can they say at all what the overall fluxes are from the soils and vegetation exchange with the atmosphere?

From the abstract: in winter. . .. Small overall Hg(0) deposition. Is this a net over the winter? i.e. the snowpack at the end of winter has more Hg than earlier in winter? What does this say about snow melt which this study seems to ignore? Are there measurements from the snow melt period? If so, they should be incorporated here so that a total "year round" net Hg deposition can be calculated.

And in spring there were AMDEs and post-AMDE re-emission. But the total net for spring was an overall loss of H(0) from the snowpack? Where did this added snowpack Hg(0) come from to be lost?

Finally, in summer, what was the overall net increase/deposition? And taken in total what were the yearly net fluxes? I feel this set of questions are important because of the still uncertain seasonal loss versus loading calculations folks have been trying to make. This study may have the most up to date information to address this need. They cite the Douglas et al., (2012) review for some mention of this (page 1, 37-38) but that paper provides a wide range of re-emission values.

---

## Referee Comment (RC3) · Anonymous Referee #3 · 16 Jul 2019

Overall, I enjoyed reading the manuscript by Jiskra et al. on "Insights from mercury stable isotopes on terrestrial – atmosphere exchange of Hg(0) in the Arctic tundra". I agreed with another review that perhaps some information and data have already been presented in previous papers by the team, but I also think this is a very nice "wrap up" paper for all these results, they are complicated and I think the authors did an excellent job to put together the story, despite with some degree of uncertainty.

I agree with most comments posted by Referee 1 & 2, I only have minor comments here and one suggestion as listed below:

P.2 L3: State percent of Hg to Arctic Ocean derived from Arctic Rivers? I thought Sonke

[Figure]

et al. (2018 PNAS) found that values. P.2 L4: Suggest ".... bioaccumulates and bio-magnifies....", without the latter, we don't have too much Hg problems. P.2 L21: Do you want to emphasize abiotically, photochemically and microbially induced re-emission of Hg(0)? How they may be distinguished by Hg isotopes? P.2 L28: Regarding to "triple isotopic fingerprint", I think we mainly rely on MDF and odd-MIF for that, less so with even-MIF, right? P.2 L29: Regarding to "60-90% of Hg in soils is derived from Hg(0) uptake by vegetation", does this already account for wet vs. dry deposition only? how about geogenic source? P.3 L24/25: State the lowest amount of Hg needed for isotopic analysis. P.3 L30/31: Not quite clear to me about "40 vol.% 2HNO3:1HCl"? P.5 L29: Typo-arCtic snow P.5 L39: Wrong unit: ~2000 ng m-2 P.6 L35: Is it correct to refer the text to Fig. 6 here? P.7 L1/2: For "...as a promising tracer to distinguish between atmospheric deposition of Hg(II) in precipitation...", do you mean to distinguish deposition of precipitation Hg(II) from gaseous Hg(0)? P.7 L31: Such large, estimated enrichment factor is interesting to see, would be interesting to propose how to "test" that experimentally. P.8 L1-10: This is cool explanation! Last suggestion: Besides summarizing better on the seasonal differences on these processes as suggested by another referee, I wonder if vegetation uptake is the dominant pathway for Hg(0) to deposit onto arctic tundra soils, should the authors consider here (or another paper) to show the global warming effects on Hg(0) deposition in longer summer in the future, and any impacts on Hg isotopic signatures in soils?

―――――――――――――――――――

---

## Referee Comment (RC4) · Anonymous Referee #4 · 17 Jul 2019

This study combines Hg fluxes and Hg isotopes in atmospheric Hg(0) and Hg(0) from snow air and soil air to investigate the fate of Hg(0) in Arctic tundra terrestrial environment. This study is part of a larger, systematic study, which I think is well planned and well carried out. The data on Hg isotopes in soil air and snow air is very novel, and they indeed provide new insight in Hg cycle in Arctic and support the conclusions from many previous studies about terrestrial-air Hg cycle that were based on experiments and field observations. The paper is also well written and the data is clearly and properly analyzed and interpreted. Overall, I recommend the publication of this paper with minor revisions:

General comment:

The observation of opposite Hg isotope signals between snow air and soil air is indeed interesting. I think it would add some value to this paper if the authors can give a more thorough thoughts on this, especailly regarding the mechanism of Hg isotope signals in snow air. The current interpretation relying on lichen uptake does not seem to be very convincing. The source and process of Hg in lichen and snow air could be very different. Are there any redox processes within snow that could produce the isotope pattern in snow air and what is the possible mechanism?

Specific comment:

1. P6, L1: it is a little confusing for the word "complementary". Does lichen and snow air represent two complementary pools of Hg? Lichen represents a long term accumulation of atmospheric Hg(0) throughout the year, whereas snow air is a more temporary pool of Hg. The source and process of Hg in lichen and snow air could be very different.

2. P6, L26-31: This section is about AMDE season, but why suddenly you switch to the discussion about May, which is after the three AMDE events? What about the isotope signals in snow air during AMDE? Should this be mentioned?

3. P6, L35: Figure 6 seems to be a wrong figure, it is not about even MIF, do you mean figure 4?

4. Figure 4: the x axis should be explained.

5. P8, L1: "do not only reflect", delete "do"

6. P8, L27: Could you show the change of MIF with Hg(0) concentration?

7. P8, L8-9: I agree, but this does not explain your high enrichment factor, which does not distinguish between pure foliar uptake and the net effect of uptake and re-emission. Re-emission would indeed affect the d202Hg and you certainly need to discuss this

scenario, but re-emission likely occurs in all situations and would not cause the difference between yours and other studies. Furthermore, re-emission is accompanied by MIF, but your data shows no change of MIF between day and night.

8. P8, L11-15: I agree with this interpretation and I believe this is a more likely interpretation than the re-emission scenario. The d202Hg in atmospheric Hg(0) is not only affected by foliar uptake. Mixing with other Hg sources should be considered in the first place. The Rayleigh model shown in Figure 6S is based on the assumption that the change of d202Hg was completely caused by processes, which should be clarified.

9. P8, L32: How did you estimate the <5% of total Hg deposition flux? Can you elaborate a little? The concentration of Hg(0) in soil air is almost lowered by half compared to atmospheric Hg(0). This seems to be a significant sink.

10. P8, L34-35: I agree that the difference between soil air and atmosphere is caused by uptake of Hg(0) by soil because the isotope signals are very consistent with the experimental work. However, the opposite Hg isotope signals between soil air and snow air do not directly support that argument that soil Hg(0) sink is minor, because the isotope signals of Hg(0) in snow air is likely controlled by other mechanisms, which I believe is not clearly identified.

---

## Referee Comment (RC5) · Anonymous Referee #5 · 22 Jul 2019

Overall, I think this is a very nice paper, certainly worthy of publication in Biogeosciences. I think the authors do mostly a good job of integrating their previous and directly related work to the results of this study, but I can possibly agree with other reviewers that it does at times come across as slightly confusing what things are new findings and what are not. That said, the paper overall hinges on very novel measurements of Hg isotopes in both snow interstitial air and soil air. It also presents some nice gradient based measurements of Hg flux and atmospheric stability, which I think do add nicely to the other parts of the paper. I do think the previous work, since it complements these new and novel measurements so well, is in the end largely written in a way that I think is entirely acceptable. If anything, the authors could perhaps go out

of their way a little more in the conclusions to more explicitly pinpoint and take credit for the particular novelty of this work in comparison to their previous work.

Specific comments:

Final paragraph of introduction: I find the write-up of these objectives miss the mark a little because they are vague. Is the purpose really just to "better understand" something or is it more pointed in trying to examine whether certain hypotheses hold up when doing some novel measurements? The list of measurements and such comes across as somewhat less focused than is actually presented. I think it is totally fine that this paper is a little descriptive, but I do think this last "purpose" paragraph could be a little more specific.

Line 36 of page 3: Is this large a variation in sample yield problematic for isotope analyses? It seems large to me, especially for mass dependent work, but if it is no issue, this could be stated here.

First half of first paragraph of section 3.1: This discussion is a little hard to follow because this study measures Hg isotope values in interstitial air, but refers to other studies that measure Hg isotopes in snow itself. Given the discussion, it seems a little unclear whether the snow interstitial air isotope signature is slightly processed (e.g., partially deposited) atmospheric mercury or is re-emitted from the snow itself into the interstitial air. I am sure this is a minor thing and just a point of clarity.

Line 35, page 6: I am unsure about the jump to referring to figure 6 here. I do not believe either of figures 4 or 5 have been introduced yet.

Line 18, page 7: Though this says Figure 1I, it looks like "figure eleven". Perhaps this could be formatted differently to avoid confusion unless this is the required convention?

Figure 2: I am unclear on whether the upper values are air above the snowpack or are indeed interstitial snow air? They appear to be above the average snow heights.

---

## Author Comment (AC1) · 12 Aug 2019

**Autor response to Reviewer comments 2, the reviewers comments are in** normal black font **and the autor response are in blue bold font.**

The manuscript is well writen, exemplarily concise and of high scientific quality. One problem is, however, that the data presented already to some degree been published in Obrist et al. (2017) doi: 10.1038/nature22997.
The present manuscript refers to this Nature paper more than 20 times, which hampering a throughout reading obtaining comprehensive informaton from text, tables and figures.

**We thank the reviewer for his positive assessment and the constructive comments.**

**The Hg0 flux dataset is the same as presented in Obrist et al. 2017. The 2017 Obrist study focused on a mass balance of Hg input vs. output in the tundra ecosystem. In this context Hg0 fluxes were presented as cumulative fluxes and Hg stable isotopes were used as source tracers. In the 2500 word letter format of Obrist et al 2017 we had to be very concise and could not discuss individual features in the dataset. In this study we re-visit the flux dataset and discuss for first time diurnal variations in fluxes and how they impact atmospheric concentrations. We also present for the first time Hg0 isotope data of interstitial snow and soil air, which was not presented in Obrist et al.**

**Based on this reviewers comment as well as other reviewers comments we will provide a clearer definition of the objectives of this study in the revised manuscript to highlight the added value of this paper compared to the Obrist et al 2017 paper as follows: "In our previous work we showed that atmospheric Hg(0) deposition to vegetation and soil represents 70% of total atmospheric deposition leading to high Hg levels in Arctic soils (Obrist et al., 2017;Olson et al., 2018). In this study we explore the use of novel mercury stable isotope measurements of Hg(0) in in interstitial snow air and soil pore air to identify the processes driving tundra Hg(0) deposition. We further discuss the effect of terrestrial-atmosphere exchange processes and planetary boundary layer stability on atmospheric concentrations and Hg stable isotope signatures of Hg(0)."**

A basic issue is that a summary tabulaton of flux and ancillary data sta9s9cs (number of observa9ons, flux data coverage (%), % of data rejected due undeveloped turbulence or fetch limita9ons etc. etc.) is missing in both papers. Please, provide a table in the main part or in a supplement.

**Mercury flux was calculated for each 30 min, so we have a total of 17568 data points for one full year (i.e., from Oct. 2015 to Sep. 2016). For the sonic data set, we have 10% of missing data (when the Monin-Obukhov length was not measured due to instrument or acquisition failure), 86% of unstable (when z/L was between −2 and −0.1), neutral (z/L between −0.2 and 0.1), and stable (z/L between 0.1 and 2) conditions that were used for the flux calculation, and 4% of very unstable/very stable conditions (z/L less than −2 and more than 2, respectively) that were removed from the data set. Besides, 92% of the Hg gradient data were correctly measured (only 8% of missing Hg concentration measurements). That means that we really calculated a Hg flux for 79% of the time.**

**A summary of this information will be added to the revised manuscript.**

The uncertainty in flux measurements is not mentoned and quantfied. Such a discussion should also include that the flux deriva9on is obtained by asynchronous Hg0 sampling of the two heights.

**Quality control of flux measurements has been discussed in Obrist et al. 2017 as follows: "For quality control, sampling lines were confirmed to be free of contamination during each field visit (approximately every six weeks, using Hg-free air; model 1100, Tekran). In addition, line intercomparisons were conducted at the same intervals to test for line biases between**

**the upper and lower inlet lines; for this, both upper and lower inlet lines were set at the same height and measurements were conducted to assess offset. Line intercomparison tests showed no substantial line offsets throughout the study, with the exception of one time when a leak was detected and immediately fixed, and fluxes before that time were corrected"**

**Gradient based measurement techniques are currently the best available method for measuring net ecosystem exchange fluxes of gaseous elemental mercury. To keep the manuscript concise, we prefer to not discuss general methodological shortcomings in this manuscript, in particular since the main objective of the study was to develop and discuss the Hg stable isotope tracer in terrestrial atmosphere exchange.**

The measured Hg0 depositon velocities should be mentoned and discussed with literature data.

**We measured net ecosystem exchange fluxes (Hg0 deposition – Hg0 re-emission), since there were no independent measurements of Hg0 re-emission we cannot calculate net Hg0 deposition or deposition velocities from our data.**

Correlaion analysis between measured gases, flux and environmental parameters is not presented.

**The focus of this study lies on discussing trends in diurnal variation and time series, were correlation analysis is not very powerful. An extensive discussion of correlations between measured gases, fluxes and environmental parameters would in our view lead a much longer manuscript. The manuscript is already quite extensive with 8 Figures and we prefer to keep it in the present length.**

To improve the readability, consider assigning the oxida9on state of Hg in delta and capital delta nota9ons (e.g. $\delta^{202}Hg^0$, $\Delta^{199}Hg^{II}$) when found appropriate. **The nomenclature used in this study is established in the Hg stable isotope community**

Page 2, Line 1: *drawn down*, consider revising **sentence revised**

Page 2, Line 14 – 15: Lindberg *et al.* 1998 is outdated (suggesting foliage as net source of $Hg^0$). Consider e.g. Bash and Miller (AE, 2009) or Castro et al. (Atmosphere, 2016) **We consider the work of Lindberg et al 1998 as pioneering and would like to give them credit for this and keep the reference. We added the two references suggested by the reviewers to the manuscript.**

Page 3, Line 16: "1.5m apparat", mistake? **Typo corrected**

Page 4, Line 25: an aerodynamic... consider the aerodynamic... **revised as suggested**

Page 4, Line 29 – 30: Φh the universal temperature profile, provide a reference for the mathematical form used. **The respective reference was added to the manuscript: Monson and Baldocchi, 2014: ISBN 978-1-107-04065-6 (Terrestrial biosphere-atmosphere fluxes. Cambridge University Press)**

Page 5, Line 21: Provide ±SD of the mean **We added the standard deviation of the flux measurements**

Page 6, Line 10: ... remained relatively low... try to be more concise (numbers) **We defined low as <0.1 ng m$^{-3}$ and adjusted the manuscript accordingly**

Page 6, Line 16: ODE's without explanation. Define Ozone depletion events as ODEs. **The definition of ODE's was added**

Page 6, Line 22:  Provide median also, if there is a substantial difference with mean **We added the median of the flux to the revised manuscript**

---

## Author Comment (AC2) · 12 Aug 2019

**Autor response to Reviewer comments 2, the reviewers comments are in** normal black font **and the autor response are in blue bold font.**

Review of Jiskra et al.

This manuscript is well written and represents the work of a strong field and laboratory team focused on mercury deposition to the Arctic. The study is well conceived and presented. It will be of interest to a variety of leadership, particularly because there is an increasing interest in understanding the source and ultimate fate of Hg in permafrost soils. I have some small to moderate comments/recommendations:

**We thank the reviewer for this thorough and constructive comments**

Title: "Insights" is not a strong word for this study. I recommend a far better title. "Mercury stable isotopes reveal XYZ on the terrestrial-atmosphere exchange of Hg(0) in arctic tundra"

**The problem with the suggestion of the reviewer is that we would have to highlight only one major finding and through this we would give the others less attention. We therefore prefer to keep the rather general title.**

This brings me to a question about the conclusions (more later): what does this study say about the seasonal net in versus out of Hg with respect to the snow pack, inferring snow melt, and summer soils? I kept hoping they would provide a seasonal diagram with the Hg % deposited, % re-emitted, and overall fluxes for their site.

**The overall fluxes and how they are distributed over the different seasons were discussed in Obrist et al. 2017. In order to avoid too much overlap between two studies (see comment to reviewer 1) we refer to the refer to the Obrist et al. study for overall mass balance.**

Page 1: 12: in arctic mercury Make sure "Arctic" versus "arctic" are correct You could say "the Arctic mercury cycle" **changed as suggested**

17-18 net emission fluxes based on the AMDEs or over the entire spring there was an overall net loss? **Net re-emission was observed in the entire spring, see discussion in section 3.2. The sentence starts with in spring, we therefore consider this statement to be clear, no changes made.**

32: Hg emission **changed as suggested**

33: such as the Arctic, through **changed as suggested**

36: (AMDEs), leads **changed as suggested**

Page 2: 32: Toolik Lake is on the Arctic Coastal Plain of Alaska. Not the Interior. **changed as suggested**

Page 3: 15: an assembly with two 47mm diameter single stage filters (?) membrane **Filter assembly is the technical term used by the manufacturer and we prefer to keep this terminology in the manuscript, no changes made (see Figure 1B in the SI)**

16: apart **changed as suggested**

18: Since the site visits were every 6-8 weeks: did the filter void spaces fill up? **Soil air lines were positioned under-ground and covered by soil. We did not inspect the filter packs**

**during site visits, to minimize disturbance of the sampling system. When the soil was saturated with water we saw a decrease in pressure/flow rate of the sampling system and manually switched off the sampling to minimize the risk of water intrusion. No changes made to the manuscript**

20: The 5.7 to 17.7 m3. Is this for the long term or short term deployments? **This information is for both, short and longterm deployments as indicated by the word overall, no changes made**

22: no comma after 22:00 **changed as suggested**

24: what is "sufficient Hg for analysis"? **information added (<2.5 ng)**

26: the soil pore **changed as suggested**

29: no comma after IAC trap **changed as suggested**

30: no comma after oven system **changed as suggested**

Page 4: 11: were determined **changed as suggested**

31: Here Teflon has the registered trademark but not earlier when "Teflon" is written (page 3, line 15) **Trademark sign added to page 3**

Page 5: 5: do you mean sonic sounder? There is little information on the meteorologi- cal measurement instrumentation except perhaps here? **Correct, we used a Metek USA-1 sonic anemometer (Metek GmbH, Elmshorn, Germany). The respective information will be added to the revised manuscript.**

Was wind direction measured and analyzed? Any association between MDEs and prior wind from the coast? Any association at all with wind direction and the Hg values measured? **We added HYSPLIT back trajectory analysis to the revised manuscript to track the origin of the air masses.**

Results and Discussion Were there any measurements during snow melt? How/why were the different time intervals selected? They seem arbitrary. Perhaps non-AMDE winter and AMDE conditions instead of winter and spring? The "spring" is actually colder than "winter." **The spring window includes snow-melt conditions, where no AMDE's were observed. We therefore prefer to keep the terminology. The observation of the reviewer is right, in winter 2015/2016 average temperatures were higher than in spring. For atmospheric mercury redox chemistry and atmospheric boundary layer stability the solar radiation is more important than absolute temperature. This situation was very unusual, normally temperatures in winter are colder and around -40°C but in this particular winter temperatures were around the freezing point for a couple of days around new year.**

Where are the data from 5 April to 3 May?

Was snow melt part of the 3 May to 9 Sep timeframe? Or an inundated tundra surface following melt?

Any relationships between summer seasonal thaw and Hg?

**The Snow melt period was included in the spring season discussion (Paragraph 3.2), which was renamed. The systematics of Hg0 isotope signatures in interstitial snow air during the snow melt**

**period are shown in Figure S3. In General it has to be recognized that during snow melt mercury is expected to be emitted to the atmosphere in pulses, which we were able to track through the flux measurements, however our Hg stable isotope sampling scheme had a too course resolution to track such short-term pulses.**

30: under the snowpack **changed as suggested**

38: coastal snowpacks **changed as suggested**

Page 6: 4: remove "also" as "possibly" is already in there **changed as suggested**

8 (Figure 3): How were AMDEs defined?

The main text Figure order is 1, 2, 8, 3, 6, 4, 5, 7 Please reorder in numerical order **Figure order was corrected**

16: similar AMDE events **changed as suggested**

8-20: Was there any analysis of the wind back trajectories or the Barrow (now UtqiaġÌĞvik)basedGMDozonetoidentifywhethertheAMDEswereregionaltothecoast? **Good suggestion, we will include backward trajectories to the revised manuscript**

Page 7: 8-9: each night, and the strongest, **changed as suggested**

24-28: Here and elsewhere where these types of data are presented. Are the different pools statistically significantly different? Providing the analytical errors is helpful but a statistical analysis of these data is in order. From a visual perspective the standard deviations likely cross over and there is no significant difference. **We agree that statistics have been missing and will add the results of statistical t-test in the revised manuscript.**

Page 8 24: data in Figure 7. Same comment as above about statistical analyses **The results of the statistical tests are provided in the main text (P7, L32 and P8, L23-24 of the Discussion version).**

39: strongly affect **changed as suggested**

Conclusions: I really like the information in this manuscript and how it is presented. There is a lot of work here.

However, the conclusions read like a summary of the results. This study could go far in identifying the seasonal aspects of Hg deposition and re-emission but the authors mostly just summarize. All the way back to the title word "insights" I recommend they go farther.

What can they say about the Hg seasonality of deposition in the Arctic?

In the introduction the authors start with Hg being a pollutant and then introduce AMDEs and talk about snowpack re-emission. A large question there is- what frac- tion of snowpack Hg makes it into runoff and of that how much ends up stored in soils?

They then mention tundra soils can draw down summertime Hg(0). So can they say at all what the overall fluxes are from the soils and vegetation exchange with the atmo- sphere?

From the abstract: in winter. . .. Small overall Hg(0) deposition. Is this a net over the winter? i.e. the snowpack at the end of winter has more Hg than earlier in winter? What does this say about snow melt which this study seems to ignore? Are there measurements from the snow melt period? If so, they should be incorporated here so that a total "year round" net Hg deposition can be calculated.

And in spring there were AMDEs and post-AMDE re-emission. But the total net for spring was an overall loss of H(0) from the snowpack? Where did this added snowpack Hg(0) come from to be lost?

Finally, in summer, what was the overall net increase/deposition? And taken in total what were the yearly net fluxes? I feel this set of questions are important because of the still uncertain seasonal loss versus loading calculations folks have been trying to make. This study may have the most up to date information to address this need. They cite the Douglas et al., (2012) review for some mention of this (page 1, 37-38) but that paper provides a wide range of re-emission values.

**In general, we agree with the reviewer about the "summary" character of the conclusion paragraph and in the revised manuscript we will provide a more concise discussion on the implications of our study also in the context of climate change. Concerning most questions raised by the reviewer here we refer to our Obrist et al. 2017, Nature study, where we discuss the ecosystem mass balance including the seasonal variation. We would also like to mention that no runoff was measured in this study, we have therefore a very limited evidence to discuss overall Hg stability in soils with respect to runoff and prefer to focus the scope of this study on terrestrial – atmosphere exchange.**

---

## Author Comment (AC3) · 12 Aug 2019

**Autor response to Reviewer comments 3, the reviewers comments are in** normal black font **and the autor response are in blue bold font.**

Overall, I enjoyed reading the manuscript by Jiskra et al. on "Insights from mercury stable isotopes on terrestrial – atmosphere exchange of Hg(0) in the Arctic tundra". I agreed with another review that perhaps some information and data have already been presented in previous papers by the team, but I also think this is a very nice "wrap up" paper for all these results, they are complicated and I think the authors did an excellent job to put together the story, despite with some degree of uncertainty. **We would like to thank reviewer 3 for this very positive assessment and his constructive comments.**

I agree with most comments posted by Referee 1 & 2, I only have minor comments here and one suggestion as listed below:

P.2 L3: State percent of Hg to Arctic Ocean derived from Arctic Rivers? I thought Sonke et al. (2018 PNAS) found that values. **It's 44-50 t/year, the number was added to the revised manuscript**

P.2 L4: Suggest ".... bioaccumulates and bio- magnifies....", without the latter, we don't have too much Hg problems. **In our understanding of the terminology, the word bioaccumulation includes bioconcentration and biomagnification (see. Alexander, D. E., Bioaccumulation, bioconcentration, biomagnification. In** *Environmental Geology***, Springer Netherlands: Dordrecht, 1999; pp 43-44.). no changes made to manuscript**

P.2 L21: Do you want to emphasize abiotically, photochemically and microbially induced re-emission of Hg(0)? How they may be distinguished by Hg isotopes? **In this introduction we want to keep the discussion simple and do not want to emphasize any particular re-emission process. We refer to the publication of Jiskra et al. 2015 with respect to the use of Hg stable isotopes to distinguish different re-emission processes in soil samples. Since in our study we did observe a depletion of Hg(0) in soil gas the re-emisstion pathways were no subject of discussion. no changes made to manuscript**

P.2 L28: Regarding to "triple isotopic fingerprint", I think we mainly rely on MDF and odd-MIF for that, less so with even-MIF, right? **No, even-MIF (e.g. $\Delta^{200}$Hg) is an important tracer for atmospheric redox processes and in contrast to odd-MIF the even-MIF signature is not subject to fractionation during post deposition processes. This is why we use a combined triple isotopic fingerprint. We refer to Enrico et al. 2016 and Obrist et al. 2017 SI for in-depth discussion. no changes made to manuscript**

P.2 L29: Regarding to "60-90% of Hg in soils is derived from Hg(0) uptake by vegetation", does this already account for wet vs. dry deposition only? how about geogenic source? **Most studies cited in this context included a potential geogenic source and the percentages provided are relative to the total Hg in soils. Note that for organic soil horizons the geogenic contribution can generally be neglected, for the mineral soils the geogenic contribution varies with bedrock and can make up for example 40 % of the total Hg in the mineral B horizons at Toolik field station (see Obrist et al. 2017 for details on source attribution). no changes made to manuscript**

P.3 L24/25: State the lowest amount of Hg needed for isotopic analysis. **>2.5 ng, this information was added, please note that this is the absolute minimum required, normally we aim to recover at least 10 ng which allows for duplicate Hg isotope analysis.**

P.3 L30/31: Not quite clear to me about "40 vol.% 2HNO3:1HCl"? **This refers to a 4.2 N HNO₃, 1.2 N HCl oxidizing acid, manuscript was changes accordingly.**

P.5 L29: Typo-arCtic snow **corrected**

P.5 L39: Wrong unit: ~2000 ng m-2 **corrected**

P.6 L35: Is it correct to refer the text to Fig. 6 here? **Thanks for spotting this error, we refer to Figure 4 here. Manuscript changed accordingly**

P.7 L1/2: For "...as a promising tracer to distinguish between atmo- spheric deposition of Hg(II) in precipitation...", do you mean to distinguish deposition of precipitation Hg(II) from gaseous Hg(0)? **No, this sentence refers to the possibility to distinguish between Hg(II) deposition and direct Hg(0) deposition. Gaseous Hg(0) is oxidixed e.g. during vegetation uptake and when analyzing e.g. a soil sample Hg originating from vegetation uptake is not present as gaseous Hg(0) anymore but as Hg(II) complexed to soils. However, this Hg inherits the Hg stable isotope fingerprint of atmospheric Hg(0), which can be distinguished from Hg(II) that was deposited through precipitation. No changes made to the manuscript.**

P.7 L31: Such large, estimated enrichment factor is interesting to see, would be interesting to propose how to "test" that experi- mentally. **We are working on this but do not want to go into detail in present manuscript.**

P.8 L1-10: This is cool explanation! **Thanks**

Last suggestion: Besides summarizing better on the seasonal differences on these processes as suggested by another ref- eree, I wonder if vegetation uptake is the dominant pathway for Hg(0) to deposit onto arctic tundra soils, should the authors consider here (or another paper) to show the global warming effects on Hg(0) deposition in longer summer in the future, and any impacts on Hg isotopic signatures in soils? **Our results do not allow direct conclusions on how climate change will impact Hg and Hg stable isotope systematics in the Arctic tundra. For the revided manuscript we will consider to address some potential implications of climate change on Arctic terrestrial-atmosphere exchange and highlight areas for further research.**

---

## Author Comment (AC4) · 12 Aug 2019

**Autor response to Reviewer comments 4, the reviewers comments are in** normal black font **and the autor response are in blue bold font.**

This study combines Hg fluxes and Hg isotopes in atmospheric Hg(0) and Hg(0) from snow air and soil air to investigate the fate of Hg(0) in Arctic tundra terrestrial environ- ment. This study is part of a larger, systematic study, which I think is well planned and well carried out. The data on Hg isotopes in soil air and snow air is very novel, and they indeed provide new insight in Hg cycle in Arctic and support the conclusions from many previous studies about terrestrial-air Hg cycle that were based on experiments and field observations. The paper is also well written and the data is clearly and prop- erly analyzed and interpreted. Overall, I recommend the publication of this paper with minor revisions: **We thank reviewer 4 for this constructive and positive assessment.**

General comment:

The observation of opposite Hg isotope signals between snow air and soil air is indeed interesting. I think it would add some value to this paper if the authors can give a more thorough thoughts on this, especailly regarding the mechanism of Hg isotope signals in snow air. The current interpretation relying on lichen uptake does not seem to be very convincing. The source and process of Hg in lichen and snow air could be very different. Are there any redox processes within snow that could produce the isotope pattern in snow air and what is the possible mechanism?

**In the revised manuscript we will provide a more nuanced discussion on Hg stable isotope systematics in soil and snow air.**

Specific comment:

1.      P6, L1: it is a little confusing for the word "complementary". Does lichen and snow air represent two complementary pools of Hg? Lichen represents a long term accumulation of atmospheric Hg(0) throughout the year, whereas snow air is a more temporary pool of Hg. The source and process of Hg in lichen and snow air could be very different.

**We agree and have revised the respective paragraph in order to provide a more accurate explanation**

2. P6, L26-31: This section is about AMDE season, but why suddenly you switch to the discussion about May, which is after the three AMDE events? What about the isotope signals in snow air during AMDE? Should this be mentioned?

**We agree that this was confusing and renamed the chapter to "spring", which includes the AMDE events and smowmelt. During AMDE's we saw a shortterm increase in the Hg(0) concentration in snow air. However, we were not able to sample specifically for Hg(0) snow air during AMDE's for Hg stable isotope analysis. The time periods were to short and with a very low sampling rate in the snow air they would not be isotopically resolvable.**

3. P6, L35: Figure 6 seems to be a wrong figure, it is not about even MIF, do you mean figure 4? **Yes, thanks for spotting this, error was corrected**

4. Figure 4: the x axis should be explained. **The x axis is explained as: Mass dependent fractionation ($\delta^{202}$Hg), which is defined in equation 1. In our view no further explanations are needed, no changes were made**

5. P8, L1: "do not only reflect", delete "do" **changed as suggested**

6. P8, L27: Could you show the change of MIF with Hg(0) concentration?

**In the manuscript we write that nighttime and daytime $\Delta^{199}$Hg values are similar, as a consequence of which there is no significant variation of $\Delta^{199}$Hg with Hg(0) concentration. For the courtesy of the reviewer we show a corresponding figure below.**

[Figure]

7. P8, L8-9: I agree, but this does not explain your high enrichment factor, which does not distinguish between pure foliar uptake and the net effect of uptake and re-emission. Re-emission would indeed affect the d202Hg and you certainly need to discuss this scenario, but re-emission likely occurs in all situations and would not cause the differ- ence between yours and other studies. Furthermore, re-emission is accompanied by MIF, but your data shows no change of MIF between day and night.

**We disagree with the suggestion that Hg re-emission occurs at all time. There is are several papers showing that net foliar Hg re-emission only occurs during daytime and during nighttime a net uptake was observed (e.g. Fu, X.; Zhu, W.; Zhang, H.; Sommar, J.; Yu, B.; Yang, X.; Wang, X.; Lin, C. J.; Feng, X., Depletion of atmospheric gaseous elemental mercury by plant uptake at Mt. Changbai, Northeast China. _Atmos. Chem. Phys._ 2016, _16_, (20), 12861-12873. or Yuan, W.; Sommar, J.; Lin, C.-J.; Wang, X.; Li, K.; Liu, Y.; Zhang, H.; Lu, Z.; Wu, C.; Feng, X., Stable isotope evidence shows re-emission of elemental mercury vapor occurring after reductive loss from foliage. _Environmental Science & Technology_ 2018.). Furthermore, the odd-mass Hg isotope signatures provide a strong indication for photochemical processes, suggestion that photoreduction is the dominant process causing foliar Hg re-emission. Therefore, we are confident in our assumption that Hg(0) re-emission predominantly occurs during daytime. As explained in the manuscript daytime deposition/re-emission processes are not expected to be traceable in atmospheric Hg concentration of stable isotope signature due to strong mixing with tropospheric air. No changes were made to the manuscript.**

8. P8, L11-15: I agree with this interpretation and I believe this is a more likely inter- pretation than the re-emission scenario. The d202Hg in atmospheric Hg(0) is not only affected by foliar uptake. Mixing with other Hg sources should be considered in the first place. The Rayleigh model

shown in Figure 6S is based on the assumption that the change of d202Hg was completely caused by processes, which should be clarified.

**We agree that the calculation of a fractionation factor for foliar Hg(0) uptake from the diurnal variation observed is only possible based on the assumption that foliar uptake if the dominant factor for the variation and that there are no other major processes or sources. We will clarify this assumption in the revised manuscript. We would like to re-emphasize that the study was conducted in the arctic tundra, hundreds of km away from the next anthropogenic Hg source. As discussed in the manuscript, the diurnal patterns in Hg concentration in relation to $CO_2$ concentration and boundary layer stability provide strong evidence that the variation observed is indeed from vegetation uptake of Hg(0) and not from different sources.**

9. P8, L32: How did you estimate the <5% of total Hg deposition flux? Can you elabo- rate a little? The concentration of Hg(0) in soil air is almost lowered by half compared to atmospheric Hg(0). This seems to be a significant sink. **We compared the soil uptake flux estimated by Obrist et al. 2014 with the net ecosystem exchange flux measured at Toolik field station. We clarified this in the revised manuscript.**

10. P8, L34-35: I agree that the difference between soil air and atmosphere is caused by uptake of Hg(0) by soil because the isotope signals are very consistent with the experimental work. However, the opposite Hg isotope signals between soil air and snow air do not directly support that argument that soil Hg(0) sink is minor, because the isotope signals of Hg(0) in snow air is likely controlled by other mechanisms, which I believe is not clearly identified. **See discussion to comment 1 above.**

---

## Author Comment (AC5) · 12 Aug 2019

Overall, I think this is a very nice paper, certainly worthy of publication in Biogeo- sciences. I think the authors do mostly a good job of integrating their previous and directly related work to the results of this study, but I can possibly agree with other reviewers that it does at times come across as slightly confusing what things are new findings and what are not. That said, the paper overall hinges on very novel measure- ments of Hg isotopes in both snow interstitial air and soil air. It also presents some nice gradient based measurements of Hg flux and atmospheric stability, which I think do add nicely to the other parts of the paper. I do think the previous work, since it complements these new and novel measurements so well, is in the end largely written in a way that I think is entirely acceptable. If anything, the authors could perhaps go out of their way a little more in the conclusions to more explicitly pinpoint and take credit for the particular novelty of this work in comparison to their previous work. **We thank the reviewer for his positive assessment re-assuring us that the overall structure of the manuscript is justified.**

Specific comments:

Final paragraph of introduction: I find the write-up of these objectives miss the mark a little because they are vague. Is the purpose really just to "better understand" some- thing or is it more pointed in trying to examine whether certain hypotheses hold up when doing some novel measurements? The list of measurements and such comes across as somewhat less focused than is actually presented. I think it is totally fine that this paper is a little descriptive, but I do think this last "purpose" paragraph could be a little more specific. **We agree with the reverwer's comment and also in response to the comments of reviewer 1 we will provide a more concise and specific description of the objectives in the introduction: "In our previous work we showed that atmospheric Hg(0) deposition to vegetation and soil represents 70% of total atmospheric deposition leading to high Hg levels in Arctic soils (Obrist et al., 2017;Olson et al., 2018). In this study we explore the use of novel mercury stable isotope measurements of Hg(0) in in interstitial snow air and soil pore air to identify the processes driving tundra Hg(0) deposition. We further discuss the effect of terrestrial-atmosphere exchange processes and planetary boundary layer stability on atmospheric concentrations and Hg stable isotope signatures of Hg(0). "**

Line 36 of page 3: Is this large a variation in sample yield problematic for isotope analyses? It seems large to me, especially for mass dependent work, but if it is no issue, this could be stated here. **In general, an incomplete sample yield can lead to mass dependent fractionation during sample pre-concentration. However, based on our data and the extensive QA/QS we have no indication of such a bias. We also would like to stress that at least part of the variation in sample yields is owed to the uncertainty in Hg concentration and cumulative flow measurements.**

First half of first paragraph of section 3.1: This discussion is a little hard to follow because this study measures Hg isotope values in interstitial air, but refers to other studies that measure Hg isotopes in snow itself. Given the discussion, it seems a little unclear whether the snow interstitial air isotope signature is slightly processed (e.g., partially deposited) atmospheric mercury or is re-emitted from the snow itself into the interstitial air. I am sure this is a minor thing and just a point of clarity. **We revised the respective paragraph**

Line 35, page 6: I am unsure about the jump to referring to figure 6 here. I do not believe either of figures 4 or 5 have been introduced yet. **This was an error, we referred to Figure 4 here. The manuscript has been changed accordingly**

Line 18, page 7: Though this says Figure 1I, it looks like "figure eleven". Perhaps this could be formatted differently to avoid confusion unless this is the required convention? **We added a space between 1 and I to avoid misinterpretation as 11.**

Figure 2: I am unclear on whether the upper values are air above the snowpack or are indeed interstitial snow air? They appear to be above the average snow heights. **Correct, the snowpack height was unusually low in the winter 2016 and the highest inlet of the snow tower (30cm) was sampling in the atmosphere over the course of the campaign as indicated by the average snow height.**